# FreeReg: Image-to-Point Cloud Registration Leveraging Pretrained Diffusion Models and Monocular Depth Estimators

**Haiping Wang**[1,*]    **Yuan Liu**[2,*]    **Bing Wang**[3]    **Yujing Sun**[2]    **Zhen Dong**[1,†]
**Wenping Wang**[4]    **Bisheng Yang**[1,†]
[1]LISMARS & Hubei Luojia Laboratory, Wuhan University    [2]The university of Hong Kong
[3]The Hong Kong Polytechnic University    [4]Texas A&M University
{hpwang,dongzhenwhu,bshyang}@whu.edu.cn    yuanly@connect.hku.hk
bingwang@polyu.edu.hk    yjsun@cs.hku.hk    wenping@tamu.edu

## Abstract

Matching cross-modality features between images and point clouds is a fundamental problem for image-to-point cloud registration. However, due to the modality difference between images and points, it is difficult to learn robust and discriminative cross-modality features by existing metric learning methods for feature matching. Instead of applying metric learning on cross-modality data, we propose to unify the modality between images and point clouds by pretrained large-scale models first, and then establish robust correspondence within the same modality. We show that the intermediate features, called diffusion features, extracted by depth-to-image diffusion models are semantically consistent between images and point clouds, which enables the building of coarse but robust cross-modality correspondences. We further extract geometric features on depth maps produced by the monocular depth estimator. By matching such geometric features, we significantly improve the accuracy of the coarse correspondences produced by diffusion features. Extensive experiments demonstrate that without any training on the I2P registration task, direct utilization of both features produces accurate image-to-point cloud registration. On three public indoor and outdoor benchmarks, the proposed method averagely achieves a $20.6\%$ improvement in Inlier Ratio, a $3.0\times$ higher Inlier Number, and a $48.6\%$ improvement in Registration Recall than existing state-of-the-arts. The code and additional results are available at https://whu-usi3dv.github.io/FreeReg/.

## 1 Introduction

Image-to-point cloud (I2P) registration requires estimating pixel-to-point correspondences between images and point clouds to estimate the SE(3) pose of the image relative to the point cloud. It is a prerequisite for many tasks such as Simultaneous Localization and Mapping (Zhu et al., 2022), 3D reconstruction (Dong et al., 2020), segmentation (Guo et al., 2020), and visual localization (Sarlin et al., 2023).

To establish pixel-to-point correspondences, we have to match features between images and point clouds. However, it is difficult to learn robust cross-modality features for images and point clouds. Most existing methods (Feng et al., 2019; Wang et al., 2021; Pham et al., 2020; Jiang & Saripalli, 2022; Li et al., 2023) resort to metric learning methods like contrastive loss, triplet loss or InfoCE loss to force the alignment between the 2D and 3D features of the same object. However, due to the inherent data disparities that images capture appearances while point clouds represent structures, directly aligning cross-modal data inevitably leads to poor convergence. Consequently, cross-modality metric learning suffers from poor feature robustness (Wang et al., 2021) and limited generalization ability (Li et al., 2023).

---

[*]Equal contribution.
[†]Corresponding Authors.

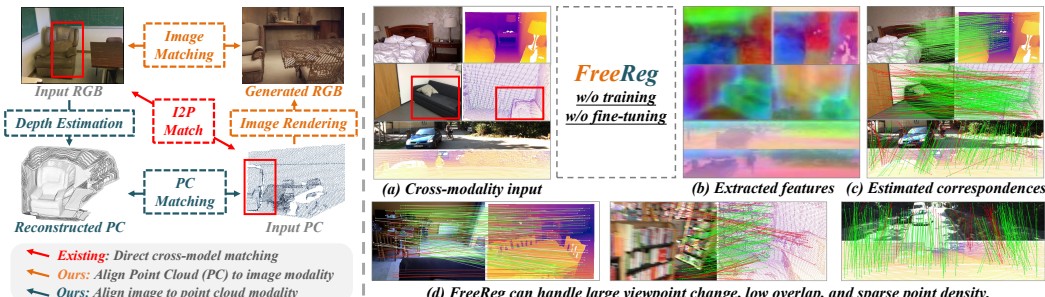

Figure 1: *Left*: FreeReg unifies the modalities of images and point clouds, which enables mono-modality matching to build cross-modality correspondences. *Right*: FreeReg does not require any training on the I2P task and is able to register RGB images to point clouds in both indoor and outdoor scenes, even for challenging cases with small overlaps, large viewpoint changes, and sparse point density.

In this paper, we propose a novel method, called *FreeReg*, to build robust cross-modality correspondences between images and point clouds with the help of recent large-scale diffusion models (Rombach et al., 2022; Zhang & Agrawala, 2023; Mou et al., 2023) and monocular depth estimators (Bhat et al., 2023; Yin et al., 2023). FreeReg avoids the difficult cross-modality metric learning and does even not require training on the I2P task. As shown in Fig. 1, the key idea is to unify the modality between images and point clouds by these large-scale pretrained models so FreeReg allows robust correspondence estimation within the same modality for cross-modality matching.

In order to convert point clouds to the image modality, a straightforward way is to project points onto an image plane to get a depth map and then convert the depth map to an image by a depth-to-image diffusion model ControlNet (Zhang & Agrawala, 2023). However, as shown in Fig. 2 (I), a depth map may correspond to multiple possible images so that the generated image from the point cloud would have a completely different appearance from the input image, which leads to incorrect matching results even with SoTA image matching methods (Sarlin et al., 2020; DeTone et al., 2018; Sun et al., 2021). To address this problem, we propose to match the semantic features between the generated images and the input image because the generated images show strong semantic consistency with the input image in spite of different appearances. Inspired by recent diffusion-based semantic correspondence estimation methods (Tang et al., 2023; Zhang et al., 2023), we utilize the intermediate feature maps in the depth-to-image ControlNet to match between depth maps and images. As shown in Fig. 2 (II), we visualize the diffusion features of the depth map and the RGB image. Then, we utilize the nearest neighbor (NN) matcher with mutual check (Wang et al., 2022a) to establish correspondences between them. We find that such semantic features show strong consistency even though they are extracted on depth maps and images separately, making it possible to build robust cross-modality correspondences. However, the semantic features are related to a large region of the image. Such a large receptive field leads to coarse-grained features and only sparse correspondences in feature matching.

We further improve the accuracy of our cross-modality correspondences with the help of the monocular depth estimators (Bhat et al., 2023). Recent progress in monocular depth estimators enables metric depth estimation on a single-view image. However, directly matching features between the point cloud and the estimated depth maps from the input image leads to poor performance as shown in Fig. 2 (III). The main reason is that the predicted depth maps are plausible but still contain large distortions in comparison with the input point cloud. The distortions prevent us from estimating robust correspondences. Though the global distortions result in noisy matches, the local geometry of the estimated depth maps still provides useful information to accurately localize keypoints and densely estimate fine-grained correspondences. Thus, we combine the local geometric features (Choy et al., 2019) extracted on the estimated depth maps with the semantic features extracted from diffusion models as the cross-modality features, which enable dense and accurate correspondence estimation between images and point clouds, as shown in Fig. 2 (IV).

In summary, FreeReg has the following characteristics. 1) FreeReg combines coarse-grained semantic features from diffusion models and fine-grained geometric features from depth maps for accurate cross-modality feature matching. 2) FreeReg does not require training on the I2P task,

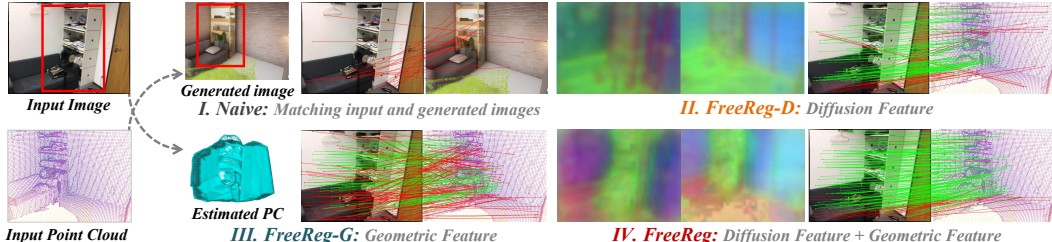

Figure 2: To unify the modalities of point clouds (PCs) and images, *I*: a straightforward way is to generate RGB images from point clouds by depth-to-image diffusion models. However, the generated images usually have large appearance differences from the query images. *II*: We find that the intermediate features of diffusion models show strong semantic consistency between RGB images and depth maps, resulting in *sparse but robust* correspondences. *III*: We further convert RGB images to point clouds by a monocular depth estimator and extract geometric features to match between the input and the generated point clouds, yielding *dense but noisy* correspondences. *IV*: We propose to fuse both types of features to build *dense and accurate* correspondences.

which avoids the unstable and notorious metric learning to align local features of point clouds and images. 3) FreeReg significantly outperforms existing fully-supervised cross-modality registration baselines (Pham et al., 2020; Li et al., 2023). Specifically, on the indoor 3DMatch and ScanNet datasets and the outdoor KITTI-DC dataset, FreeReg roughly achieves over 20% improvement in Inlier Ratio, a 3.0× more Inlier Number, and a 48.6% improvement in Registration Recall.

## 2 RELATED WORK

**Image-to-point cloud registration.** In order to establish correspondences between images and point clouds for pose recovery, most existing methods (Li et al., 2015; Xing et al., 2018; Feng et al., 2019; Lai et al., 2021; Wang et al., 2021; Pham et al., 2020; Liu et al., 2020; Jiang & Saripalli, 2022; Li et al., 2023) rely on metric learning to align local features of images and point clouds (Feng et al., 2019; Pham et al., 2020; Lai et al., 2021; Jiang & Saripalli, 2022), or depth maps (Liu et al., 2020; Wang et al., 2021; Li et al., 2023). However, these methods often require cross-modal registration training data (Pham et al., 2020; Wang et al., 2021; Jiang & Saripalli, 2022; Li et al., 2023; Kim et al., 2023) and show limited generalization ability (Pham et al., 2020; Wang et al., 2021; Li et al., 2023; Ren et al., 2022; Yao et al., 2023) due to the difficulty in the cross-modality metric learning. In contrast, FreeReg does not require any training and fine-tuning on the I2P registration task and exhibits strong generalization ability to both indoor and outdoor scenes.

Some other methods directly solve image-to-point cloud registration as an optimization problem (David et al., 2004; Campbell et al., 2019; Arar et al., 2020; Wang et al., 2023a; Zhou et al., 2023), which regresses poses by progressively aligning keypoints (Li & Lee, 2021; Ren et al., 2022; Campbell et al., 2019), pole structures (Wang et al., 2022b), semantic boundaries (Liao et al., 2023), or cost volumes (Wang et al., 2023a) of RGB images and depth maps. However, these methods heavily rely on an accurate initial pose (Wang et al., 2021; Liao et al., 2023) to escape from local minima in optimizations. FreeReg does not require such a strictly accurate initialization because FreeReg matches features to build correspondences to handle large pose changes.

**Diffusion feature extraction.** Recently, a category of research (Ho et al., 2020; Song et al., 2020a;b; Karras et al., 2022; Song & Ermon, 2019; Dhariwal & Nichol, 2021; Liu et al., 2023), known as diffusion models, has demonstrated impressive generative capabilities. Based on that, with the advent of classifier-free guidance (Ho & Salimans, 2022) and billions of text-to-image training data (Schuhmann et al., 2022), a latent diffusion model, specifically stable diffusion (Rombach et al., 2022), has shown remarkable text-to-image generation capabilities. Building upon this, existing methods have demonstrated the exceptional performance of Stable Diffusion internal representations (Diffusion Feature) (Kwon et al., 2022; Tumanyan et al., 2023) in various domains such as segmentation (Amit et al., 2021; Baranchuk et al., 2021; Chen et al., 2022b; Jiang et al., 2018; Tan et al., 2022; Wolleb et al., 2022), detection (Chen et al., 2022a), depth estimation (Duan et al., 2023; Saxena et al., 2023b;a). These methods only extract diffusion features on RGB images utilizing Stable Diffsuion. Our method extracts diffusion features on RGB and depth maps based on recent finetuned diffusion

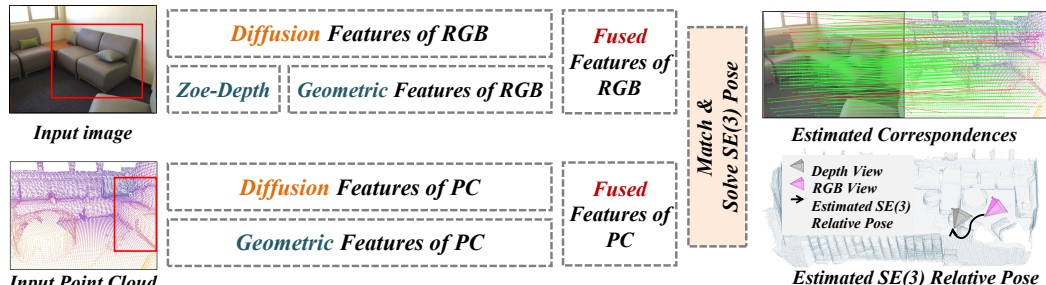

Figure 3: *FreeReg pipeline.* Given a point cloud (PC) and a partially overlapping RGB image, FreeReg extracts diffusion features and geometric features for the point cloud and the image. These two features are fused and matched to establish pixel-to-point correspondences, on which we compute the SE(3) relative pose between the image and the point cloud.

models ControlNet (Zhang & Agrawala, 2023) or T2IAdaptor (Mou et al., 2023), which efficiently leverage depth, semantic maps, and sketches to guide stable diffusion in image generation.

**Diffusion feature for matching.** Some recent works utilize diffusion features for representation learning (Kwon et al., 2022) and semantic matching (Luo et al., 2023a; Tang et al., 2023; Hedlin et al., 2023; Zhang et al., 2023) among RGB images capturing objects across instances and categories. In comparison, our method shows the effectiveness of diffusion features in learning cross-modality features for image-to-point cloud registration.

**Monocular depth estimator** Monocular depth estimation inherently suffers from scale ambiguity (Chen et al., 2016; 2020; Xian et al., 2018; 2020). With more and more monocular depth training data (Guizilini et al., 2023; Antequera et al., 2020; Wilson et al., 2023), recent works (Bhat et al., 2021; 2022; Jun et al., 2022; Li et al., 2022; Yang et al., 2021; Yin et al., 2021; 2019; Yuan et al., 2022; Guizilini et al., 2023; Yin et al., 2023) learn scene priors to regress depth values in real metric space and show impressive results. We employ a SoTA metric depth estimator Zoe-Depth (Bhat et al., 2023) to recover point clouds in the same metrics corresponding to the RGB images.

## 3 METHOD

Let $I \in \mathbb{R}^{H \times W \times 3}$ be an RGB image and $P \in \mathbb{R}^{N \times 3}$ be a point cloud. We first project $P$ to a depth map $D \in \mathbb{R}^{H' \times W'}$ on a camera pose, which is calculated from the depth or LiDAR sensor center and orientation. More details about this projection are given in the supplementary material. FreeReg aims to match the cross-modality features extracted on $I$ and $D$ to establish correspondences and solve the relative pose between them. The pipeline of FreeReg is illustrated in Fig. 3. Specifically, We extract diffusion features (Sec. 3.2) and geometric features (Sec. 3.3) for feature matching and then estimate the I2P transformation estimation from the matching results. We begin with a brief review of diffusion methods, which we utilize to extract cross-modality features.

### 3.1 PRELIMINARY: STABLE DIFFUSION AND CONTROLNET

The proposed cross-modality features are based on ControlNet (Zhang & Agrawala, 2023) (CN) so we briefly review the related details of ControlNet in this section. Diffusion models contain a forward process and a reverse process, both of which are Markov chains. The forward process gradually adds noise to the input image in many steps and finally results in pure structure-less noise. The corresponding reverse process gradually denoises the noise step-by-step to gradually recover the structure and generate the image. Stable Diffusion (Rombach et al., 2022) (SD) is a widely-used diffusion model mainly consisting of a UNet which takes noisy RGB images as input and predicts the noise. The original Diffusion model only allows text-to-image generation. Recent ControlNet (Zhang & Agrawala, 2023), as shown in Fig. 4 (b), adds an additional encoder to process depth maps and utilizes the extracted depth features to guide the reverse process of SD, enabling SD to generate images coherent to the input depth map from a pure Gaussian noise. In FreeReg, we utilize CN and SD to extract cross-modality features for feature matching.

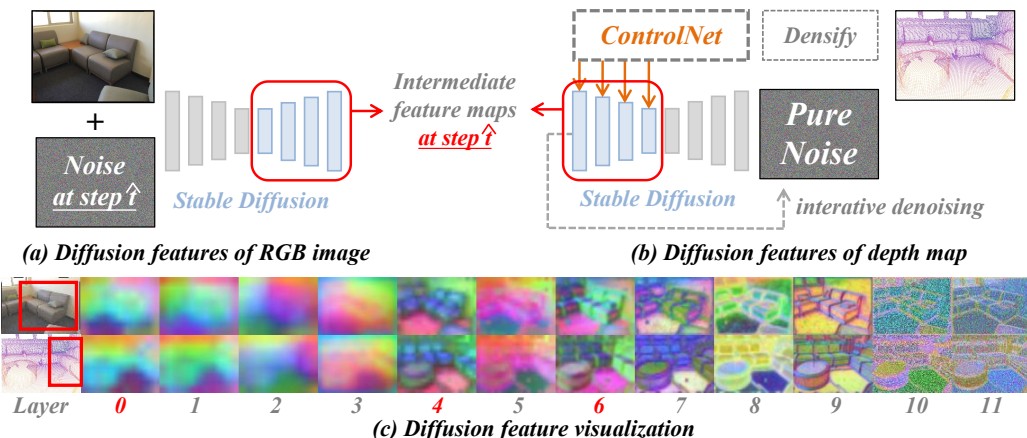

Figure 4: *Diffusion feature extraction* on (a) images and (b) depth maps. (c) Visualization of diffusion features.

## 3.2 DIFFUSION FEATURES ON CROSS-MODALITY DATA

Directly generating an image from the input depth map suffers from appearance inconsistency with the input image, which results in inaccurate feature matching. Instead of generating an explicit image, we resort to the intermediate feature maps of stable diffusion models for cross-modality feature matching. The overview is shown in Fig. 4.

**RGB diffusion feature.** As shown in Fig. 4(a), we perform the forward process of SD (Rombach et al., 2022) to add noise to the input RGB image, which results in a noisy image on a predefined step $\hat{t}$. The noisy image is fed to the UNet of the SD and the intermediate feature maps of the UNet decoder are used as the diffusion feature for the input RGB image.

**Depth diffusion feature.** Given the depth maps, we first densify them using traditional erosion and dilation operations (Ku et al., 2018). As shown in Fig. 4 (b), we propose to feed the depth map to a CN (Zhang & Agrawala, 2023) as a condition to guide the reverse process of SD. With such a condition, SD gradually denoise a pure Gaussian noise until the same predefined step $\hat{t}$ and then we use the feature maps in the SD UNet decoder as the depth diffusion features. An alternative way is to directly treat the depth map as an RGB image for diffusion feature extraction, which however leads to poor performance as shown in the supplementary material.

**Layer selection.** The remaining problem is about which layer to be used for feature extraction. Visualization of extracted diffusion features on RGB images and depth maps are given in Fig. 4(c). It can be observed that the features of early upsampling layers with layer index $l \leq 6$ show strong consistency between RGB and depth data. Features of later upsampling layers with an index larger than 6 show more fine-grained details like textures that no longer exhibit consistency. Therefore, we use features of early layers 0,4,6 as our diffusion features. To reduce the feature dimension on each layer, we apply a Principal Component Analysis (PCA) to reduce the feature dimension to 128. The resulting diffusion features of RGB image $I$ and depth map $D$ are $F_d^I$ and $F_d^D$ respectively, both of which are obtained by concatenating the features from different layers and L2 normalized.

## 3.3 GEOMETRIC FEATURES ON CROSS-MODALITY DATA

The above diffusion feature is extracted from a large region on the image, which struggles to capture fine-grained local details and estimates only sparse correspondences as shown in Fig. 5 (b/e). To improve the accuracy of these correspondences, we introduce a so-called geometric feature, leveraging the monocular depth estimator Zoe-Depth (Bhat et al., 2023).

Specifically, we utilize Zoe-Depth to generate per-pixel depth $D^Z$ for the input RGB image $I$ and recover a point cloud from the generated depth map. Then, we employ a pre-trained point cloud feature extractor FCGF (Choy et al., 2019) to extract per-point features, which serve as the geometric features of their corresponding pixels in the image $I$. We construct geometric features for pixels of the depth map $D$ in the same way. As illustrated in Fig. 5 (c/f), solely matching geometric features produces many outlier correspondences due to large distortion in the single-view depth

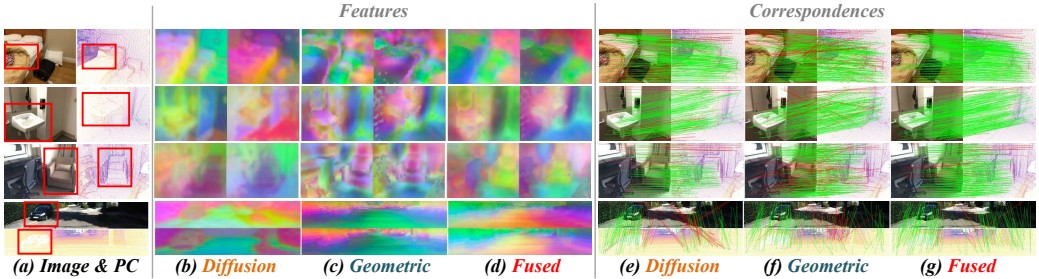

Figure 5: *Visualization of features and estimated correspondences*. (a) Input images and point clouds. (b), (c), and (d) show the visualization of diffusion, geometric, and fused feature maps respectively. (e), (f), and (g) show the pixel-to-point correspondences estimated by the nearest neighbor (NN) matcher using diffusion, geometric, and fused features respectively. Diffusion features estimate reliable but sparse correspondences. Geometric features yield dense matches but with more outliers. Fused features strike a balance between accuracy and preserving fine-grained details, resulting in accurate and dense matches.

estimation. However, these geometric features provide local descriptions of the geometry, which are more localized and enable more accurate correspondence in cooperation with the diffusion features.

## 3.4 FUSE BOTH FEATURES FOR I2P REGISTRATION

**Fuse features.** In this section, we propose to fuse the two types of features to enable accurate correspondence estimation, as shown in Fig. 5. Note that we uniformly sample a dense grid of keypoints on both the depth map and the image. Then, we extract the above diffusion features and geometric features on the keypoints. Both features are normalized by their L2 norm before the fusion. Specifically, we follow (Zhang et al., 2023) to fuse two kinds of features on each keypoint in $I$ or $D$ by

$$F = [wF_d, (1-w)F_g], \tag{1}$$

$w$ is a fusion weight, $[\cdot, \cdot]$ means concatenating on feature dimension, and $F$ is the resulting FreeReg feature.

**Pixel-to-point correspondences.** Given two sets of fused features $F^I$ on RGB image $I$ and $F^D$ on depth map $D$, we conduct nearest neighborhood (NN) matching with a mutual nearest check (Wang et al., 2022a) to find a set of putative correspondences. Note that the pixel from the depth map $D$ in each match corresponds to a 3D point in point cloud $P$.

**Image-to-point cloud registration.** To solve SE(3) poses of RGB image $I$ relative to $P$. A typical approach is to conduct the Perspective-n-Point (PnP) algorithm (Lepetit et al., 2009) on the established pixel-to-point correspondences. However, we have estimated a depth map corresponding to RGB using Zoe-Depth (Bhat et al., 2023). Thus, we can convert the pixel-to-point correspondences to 3D point-to-point correspondences, and estimate the SE(3) relative pose using the Kabsch algorithm (Kabsch, 1976). In the supplementary material, we empirically show that using the PnP algorithm leads to a more accurate pose estimation but fails in many cases, while the Kabsch algorithm works in more cases but the estimated transformations exhibit larger errors.

## 4 EXPERIMENTS

### 4.1 EXPERIMENTAL PROTOCOL

**Datasets.** We evaluate the proposed method on three widely used datasets: (1) The *3DMatch* (Zeng et al., 2017) testset comprises RGB images and point clouds (called *I2P pairs*) from 8 indoor scenes. The point clouds used here are collected by an Asus Xtion depth sensor. We manually exclude the I2P pairs with very small overlaps resulting in 1210 I2P pairs with over $30\%$ overlaps. (2) The *ScanNet* (Dai et al., 2017) testset consists of 4,660 I2P pairs from 31 indoor scenes with more than $30\%$ overlap. To further increase the difficulty, we downsampled the input point clouds using a voxel size of 3cm, which leads to highly sparse point clouds. (3) The *Kitti-DC* (Uhrig et al., 2017) testset has 342 I2P pairs from 4 selected outdoor scenes. The sparse point clouds come from a 64-line LiDAR scan. The distance between each I2P pair is less than 10 meters.

Table 1: *Cross-modality registration performance of different methods.* "InvCP." means Inverse Camera Projection (Li & Lee, 2021).

| *Method* *SE(3) Solver* | | LCD *PnP* | SG *PnP* | DeepI2P *InvCP.* | CN+SG *PnP* | I2P-Matr *PnP* | FreeReg-D *PnP* | FreeReg-G *Kabsch* | FreeReg *PnP* | FreeReg *Kabsch* |
|---|---|---|---|---|---|---|---|---|---|---|
| **3DMatch** | *FMR(%)* | 40.1 | 50.3 | / | 64.7 | 90.6 | 91.9 | 90.7 | **94.6** | **94.6** |
| | *IR(%)* | 35.1 | 11.1 | / | 18.4 | 24.9 | 39.6 | 31.4 | **47.0** | **47.0** |
| | *IN(#)* | 4.3 | 3.1 | / | 10.9 | 49.0 | 60.8 | 49.4 | **82.8** | **82.8** |
| | *RR(%)* | / | 1.8 | / | 6.5 | 28.2 | 33.2 | 50.4 | 40.0 | **63.8** |
| **ScanNet** | *FMR(%)* | 55.1 | 53.2 | / | 64.1 | 87.0 | 95.3 | 96.4 | **98.5** | **98.5** |
| | *IR(%)* | 30.7 | 13.4 | / | 18.3 | 14.3 | 45.7 | 40.5 | **56.8** | **56.8** |
| | *IN(#)* | 5.0 | 4.7 | / | 9.1 | 24.8 | 61.5 | 84.5 | **114.4** | **114.4** |
| | *RR(%)* | / | 1.2 | / | 5.5 | 8.5 | 42.3 | 69.4 | 57.6 | **78.0** |
| **Kitti-DC** | *FMR(%)* | / | 73.4 | / | 94.2 | / | **100.0** | 94.4 | 99.7 | 99.7 |
| | *IR(%)* | / | 18.1 | / | 34.4 | / | **59.4** | 41.2 | 58.3 | 58.3 |
| | *IN(#)* | / | 12.6 | / | 51.1 | / | 103.6 | 93.6 | **132.9** | **132.9** |
| | *RR(%)* | / | 8.2 | 20.9 | 20.4 | / | 68.1 | 43.3 | **70.5** | 67.5 |

**Metrics**. Following (Choy et al., 2019; Wang et al., 2023c;b), we adopt four evaluation metrics: (1) *Feature Matching Recall (FMR)* is the fraction of I2P pairs with more than $5\%$ correct estimated correspondences. A correspondence is regarded as correctly matched if its ground truth 3D distance is smaller than $\tau_c$. $\tau_c$ is set to 0.3m for 3DMatch/ScanNet and 3m for Kitti-DC. (2) *Inlier Ratio (IR)* is the average correct correspondence proportions among all I2P pairs. (3) *Inlier Number (IN)* is the average number of correct correspondences on each I2P pair. and (4) *Registration Recall (RR)* is the percentage of correctly-aligned I2P pairs with rotation and translation errors less than $\tau_R$ and $\tau_t$ respectively. $(\tau_R, \tau_t)$ is set to $(20°, 0.5m)$ for 3DMatch/ScanNet and $(10°, 3m)$ for Kitti-DC. We provide additional results under different threshold conditions in the supplementary material.

**Baselines**. We compare FreeReg with fully supervised registration baselines. The image registration method SuperGlue (SG) (Sarlin et al., 2020) is modified to match RGB images and point clouds. LCD (Pham et al., 2020) learns to construct I2P cross-modality descriptors utilizing metric learning. DeepI2P (Li & Lee, 2021) resolve I2P registration by optimizing an accurate initial pose. We implement a cross-modality feature extraction method I2P-Matr following a concurrent work 2D3D-Matr (Li et al., 2023), where the official codes are not released yet. Meanwhile, we compare FreeReg with P2-Net (Wang et al., 2021) and 2D3D-Matr (Li et al., 2023) under their experimental protocol (Li et al., 2023) in the supplementary material, where FreeReg also achieves the best registration performance. We also adopt a baseline as mentioned in Fig. 2 (I) that first utilizes ControlNet (Zhang & Agrawala, 2023) to generate an RGB image from the target point cloud and then conducts SuperGlue (Sarlin et al., 2020) to match the input and the generated image (CN+SG). For our method, we report the results using only the diffusion feature (FreeReg-D, i.e. $w = 1$), only the geometric feature (FreeReg-G, i.e. $w = 0$), and the fused feature (FreeReg, i.e. $w = 0.5$ by default) for matching. More implementation details and analysis are provided in the supplementary material.

## 4.2 RESULTS ON THREE BENCHMARKS

The quantitative results of FreeReg and baselines on the three cross-modality registration benchmarks are given in Table 1. Some quantitative results are shown in Fig. 6.

**Correspondence quality** is reflected by *FMR*, *IR*, and *IN*. For LCD and I2P-Matr, utilizing a metric learning method to directly align cross-modality features leads to poor performance. CN+SG suffers from the appearance difference between generated images and the input images and thus fails to build reliable correspondences. For FreeReg, using solely diffusion features (FreeReg-D) or geometric features (FreeReg-G) can already yield results superior to the baselines. Utilizing both features, FreeReg achieves the best correspondence quality and outperforms baselines by a large margin with $54.0\%$ in *FMR*, $20.6\%$ in *IR*, and a $3.0\times$ higher *IN*. Note that, unlike baseline methods, FreeReg does not even train on the I2P task.

**Registration quality** is indicated by *RR*. Benefited by the high-quality correspondences, FreeReg significantly outperforms the baseline methods by a $48.6\%$ RR and FreeReg-D/G by a $22.9\%/16.4\%$ RR. Moreover, FreeReg utilizing Kabsch significantly surpasses PnP on indoor 3DMatch/ScanNet but is $3\%$ lower than PnP on the outdoor Kitti-DC. The main reason is that Zoe-Depth performs better on these two indoor datasets with an average 0.27m error but worse on the KITTI with an

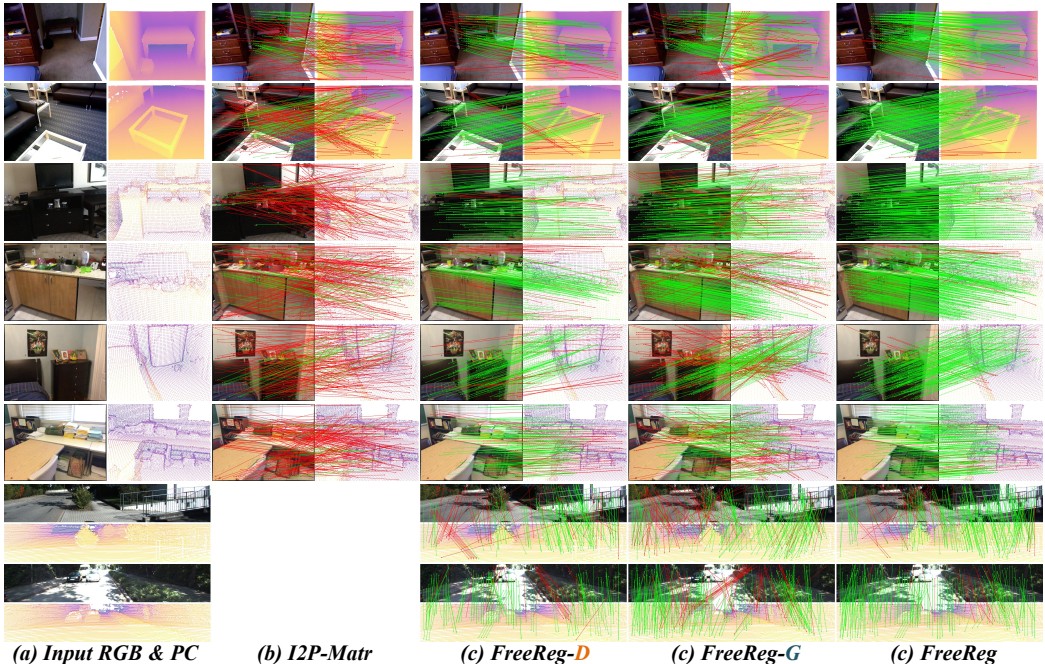

| (a) Input RGB & PC | (b) I2P-Matr | (c) FreeReg-D | (c) FreeReg-G | (c) FreeReg |

Figure 6: *Visualization of correspondences.* (a) Input RGB images and point clouds for registration. (b) Estimated correspondences from I2P-Matr. (c / d / e) Estimated correspondences from FreeReg-D / FreeReg-G / FreeReg.

average 3.4m error. In the supplementary material, we further provide more analysis and find that PnP achieves more accurate results while Kabsch provides plausible results in more cases.

## 4.3 MORE ANALYSIS

We conduct comprehensive ablation experiments on FreeReg in the following. More analysis about how to fine-tune FreeReg for performance gains, runtime analysis and acceleration strategies of FreeReg, FreeReg performance under different thresholds, more qualitative comparisons and ablation studies are provided in the supplementary material.

### 4.3.1 ABLATING DIFFUSION FEATURE EXTRACTION

In this section, we evaluate on a validation scene "bundlefusion-office0" (BFO) which is not included in the testset to tune hyperparameters in diffusion feature layer selection and diffusion step $\hat{t}$ selection. Subsequently, we report their performances on the 3DMatch dataset.

**Diffusion layer selection**. In table 2 (a-i), we report the size of output feature maps of 8 layers in the UNet of Stable Diffusion. The feature map size is divided into three levels, i.e. small group ($8 \times 11$, layers 0-2), medium group ($16 \times 22$, layers 3-5), and large group ($32 \times 44$, layers 6-8). We select the layers with the best registration performance and reasonable matching quality from each level on BFO, specifically layers 0, 4, and 6, to construct our Diffusion Features. Then, in table 2 (j-m), we ablate the layer selection in constructing diffusion features. It can be seen that concatenating features of 0,4,6 layers significantly improves the correspondence quality and registration performance. The results from 3DMatch further validate the effectiveness of our choice. More ablation studies on the diffusion layer selection are provided in the supplementary material.

**Diffusion step selection**. In Table 3, we aim to determine the diffusion step $\hat{t}$. The experimental results demonstrate that the Diffusion Features from $\hat{t} = 150$ achieve the best registration performance on BFO. Results on 3DMatch confirm its effectiveness.

### 4.3.2 ABLATING FEATURE FUSION WEIGHT

Table 2: *Layer selection in diffusion feature extraction*. "Feature map" means the size of the feature map in the form of "channel×width×length".

| ID | Layer | Feature Map (channel×h × w) | BFO FMR(%) | IR(%) | IN(#) | RR(%) | 3DMatch FMR(%) | IR(%) | IN(#) | RR(%) |
|---|---|---|---|---|---|---|---|---|---|---|
| (a) | 0 | $1280 \times 8 \times 11$ | 88.9 | 42.7 | 18.9 | 14.4 | 89.5 | 39.7 | 17.6 | 16.7 |
| (b) | 1 | $1280 \times 8 \times 11$ | 91.5 | 42.1 | 19.1 | 12.4 | 86.9 | 39.7 | 18.1 | 15.8 |
| (c) | 2 | $1280 \times 8 \times 11$ | 86.3 | 42.9 | 21.2 | 14.4 | 84.2 | 39.7 | 20.2 | 16.9 |
| (d) | 3 | $1280 \times 16 \times 22$ | 87.6 | 42.7 | 45.9 | 23.5 | 88.4 | 41.0 | 47.2 | 23.0 |
| (e) | 4 | $1280 \times 16 \times 22$ | 91.5 | 36.0 | 31.7 | 24.2 | 92.1 | 35.3 | 32.9 | 26.0 |
| (f) | 5 | $1280 \times 16 \times 22$ | 89.5 | 35.4 | 28.1 | 22.9 | 91.7 | 35.5 | 28.9 | 25.6 |
| (g) | 6 | $1280 \times 32 \times 44$ | 92.8 | 31.3 | 45.5 | 30.1 | 89.4 | 31.4 | 51.7 | 28.7 |
| (h) | 7 | $640 \times 32 \times 44$ | 90.8 | 19.9 | 34.3 | 14.4 | 85.2 | 19.6 | 35.1 | 22.1 |
| (i) | 8 | $640 \times 32 \times 44$ | 88.9 | 17.2 | 28.4 | 9.8 | 82.9 | 16.8 | 27.5 | 17.3 |
| (j) | [0,4] | $256 \times 32 \times 44$ | 93.5 | **44.6** | 25.9 | 25.5 | **92.5** | 41.3 | 34.7 | 26.5 |
| (k) | [0,6] | $256 \times 32 \times 44$ | 92.8 | 40.2 | 53.9 | 34.0 | 91.4 | 38.5 | **62.2** | 32.9 |
| (l) | [4,6] | $256 \times 32 \times 44$ | 91.5 | 36.4 | 45.8 | 32.0 | 91.4 | 35.6 | 56.2 | 30.7 |
| (m) | [0,4,6] | $384 \times 32 \times 44$ | **94.8** | 42.3 | **58.2** | **35.9** | 91.9 | 39.6 | 60.8 | **33.2** |

Table 3: Determining $\hat{t}$ in diffusion feature extraction.

| $\hat{t}$ | BFO FMR(%) | IR(%) | IN(#) | RR(%) | 3DMatch FMR(%) | IR(%) | IN(#) | RR(%) |
|---|---|---|---|---|---|---|---|---|
| 300 | 94.1 | 40.0 | 55.8 | 33.3 | 91.7 | 39.4 | 60.4 | 31.4 |
| 200 | 92.8 | 41.0 | 58.1 | 35.3 | 91.8 | 39.8 | 61.4 | 31.2 |
| 150 | 94.8 | 42.3 | 58.2 | **35.9** | 91.9 | 39.6 | 60.8 | **33.2** |
| 100 | 92.8 | 41.3 | 57.3 | 35.3 | 91.8 | 38.8 | 59.3 | 31.6 |
| 50 | 92.8 | 40.0 | 54.6 | 32.7 | 92.0 | 38.1 | 57.3 | 30.6 |

We ablate the fusion weight $w$ to fuse diffusion and geometric features in Table. 4 based on the baseline model FreeReg. It can be seen that FreeReg achieves the best registration performance when $w$ is set to 0.5. Moreover, we find that relying more on diffusion features, i.e., $w = 0.6$ achieves a much similar result to the default FreeReg. While relying more on geometric features, i.e., $w = 0.4$ causes a sharp performance drop of a $8.7\%$ lower IR and a $2.5\%$ lower RR. This demonstrates the robustness of the proposed diffusion features.

Table 4: Determining the fusion weight to fuse diffusion and geometric features.

| $w$ | FMR(%) | IR(%) | IN(#) | RR(%) |
|---|---|---|---|---|
| 1.0 | 91.9 | 39.6 | 60.8 | 52.6 |
| 0.7 | 94.8 | 45.1 | 74.1 | 58.5 |
| 0.6 | **95.3** | **47.1** | 81.7 | 62.3 |
| 0.5 | 94.6 | 47.0 | **82.8** | **63.8** |
| 0.4 | 93.8 | 42.9 | 73.5 | 60.3 |
| 0.3 | 91.8 | 37.5 | 61.9 | 56.5 |
| 0.0 | 90.7 | 31.4 | 49.4 | 50.4 |

## 4.4 LIMITATIONS

The main limitation is that FreeReg requires about 9.3s and 12.7G GPU memory to match a single I2P pair on a 4090 GPU, yielding much higher RR but longer time usage than baselines LCD (0.6s, 3.5G, I2P-Matr (1.7s, 2.7G), and CN+SG (6.4s, 11.6G). The reason is that we need to run multiple backward process steps of ControlNet to denoise the pure noises to reach a specific step $\hat{t}$ for feature extraction. In the supplementary material, we show how to accelerate FreeReg by $\sim 50\%$ with only a $\sim 1.4\%$ RR drop. Meanwhile, though we show the superior performance of using diffusion features for I2P registration, we manually select layers and denoising steps in the diffusion feature extraction, which could be improved by future works to automatically select good features.

## 5 CONCLUSION

We propose an I2P registration framework called FreeReg. The key idea of FreeReg is the utilization of diffusion models and monocular depth estimators for cross-modality feature extraction. Specifically, we leverage the intermediate representations of diffusion models to construct multi-modal diffusion features that show strong consistency across RGB images and depth maps. We further introduce so-called geometric features to capture distinct local geometric details on RGB images and depth maps. Extensive experiments demonstrate that FreeReg shows strong generalization and robustness in the I2P task. Without any training or fine-tuning on I2P registration task, FreeReg achieves a $20.6\%$ improvement in Inlier Ratio, a $3.0\times$ higher Inlier Number, and a $48.6\%$ improvement in Registration Recall on three public indoor and outdoor benchmarks.

## 6 Acknowledgement

This research is jointly sponsored by the National Key Research and Development Program of China (No.2022YFB3904102), the Open Fund of Hubei Luojia Laboratory (No. 2201000054), the National Natural Science Foundation of China (No.42301520), and the Innovation and Technology Commission of the HKSAR Government under the InnoHK initiative and Ref. T45-205/21-N of Hong Kong RGC.

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

# A APPENDIX

## A.1 IMPLEMENTATION DETAILS OF FREEREG

### A.1.1 DEPTH PROJECTION AND DENSIFY

We project the point cloud to a given camera pose following a classic method (Forsyth & Ponce, 2002) and set null pixels to zeros. We adopt a CPU-based depth completion method IDC-DC (Ku et al., 2018) to densify sparse depth maps. We adopt $fill\_in\_fast$ method with a kernel of $DIAMOND\_KERNEL\_7$ for ScanNet data and $fill\_in\_multiscale$ method with the default settings for Kitti-DC.

### A.1.2 DIFFUSION FEATURE

**Depth.** We adopt ControlNet (Zhang & Agrawala, 2023) for diffusion feature extraction on depth maps. The depth maps are normalized to $0 \sim 255$ following (Zhang & Agrawala, 2023). We use the pre-trained model of ControlNet conditioning on depth maps[1]. We use DDIM Sampling strategy (Song et al., 2020a) with diffusion steps set to $range(begin = 1000, end = 0, step = -50)$ and $ddim\_eta$ set to 1.0. We follow (Zhang & Agrawala, 2023) to adopt Classifier-Free Guidance (Ho & Salimans, 2022)

$$\epsilon^t = (\omega + 1)\mathcal{U}(z^t, t, \mathcal{C}) - \omega\mathcal{U}(z^t, t, \mathcal{C}_u), \tag{2}$$

where $\epsilon$ is the estimated noise at step $t^t$, $\omega$ is so-called unconditional guidance scale, $\mathcal{C}$ is the text-prompt for conditional sampling and $\mathcal{C}_u$ is the text-prompt for unconditional sampling. We set $\omega$ to 4.0 in implementation. We use depth maps as conditions for both conditional sampling process $\mathcal{U}(z^t, t, \mathcal{C})$ and unconditional sampling process $\mathcal{U}(z^t, t, \mathcal{C}_u)$. $\mathcal{C}$ is "*best quality, a photo of a room, furniture, household items*" for indoor 3DMatch and ScanNet datasets, "*a vehicle camera photo of street view, trees, cars, people, house, road, sky*" for outdoor Kitti-DC. $\mathcal{C}_u$ is set to "*lowres, bad anatomy, bad hands, cropped, worst quality*". We set $\sigma_l$ to 1.0 for all SD UNet at the step $\hat{t} = 150$.

**RGB.** We use the pre-trained Stable Diffusion v1.5 model [2] for diffusion feature extraction on RGB images. The image is normalized to $-1 \sim 1$ before being fed to SD, same as (Rombach et al., 2022; Zhang & Agrawala, 2023). We use the internal features of SD U-net at step $\hat{t} = 150$.

### A.1.3 ZOE-DEPTH

We adopt Zoe-Depth (Bhat et al., 2023) for monocular metric depth estimation. We utilized the Zoe-Depth model pretrained on the M12+NYU-Depth-v2[3] for the indoor 3DMatch and ScanNet datasets. We use the Zoe-Depth model pretrained on M12+NYU-Depth-v2+KITTI[4] for the outdoor Kitti-DC dataset. Note that the training set has no overlap with our testsets.

### A.1.4 FCGF

We use FCGF (Choy et al., 2019) for 3D geometric feature extraction on point clouds recovered from Zoe-Depth outputs or input point clouds. We use the FCGF (Choy et al., 2019) model pretrained on the 3DMatch (Zeng et al., 2017)[5] training set for the indoor 3DMatch and ScanNet datasets and the model pretrained on KITTI (Uhrig et al., 2017) [6] for the outdoor Kitti-DC dataset.

Following (Choy et al., 2019), we downsample point clouds by a voxel size of 0.025m for indoor datasets and 0.3m for outdoor datasets before FCGF extraction. However, since FCGF is constructed on downsampled point clouds, not every pixel in the depth maps has a corresponding FCGF feature. Therefore, for a query pixel, we first project it into 3D space and retrieve its nearest FCGF feature in 3D space as its FCGF feature. However, if the distance of this point to its nearest FCGF feature

---

[1]`github.com/lllyasviel/ControlNet-v1-1-nightly`

[2]`huggingface.co/runwayml/stable-diffusion-v1-5`

[3]`github.com/isl-org/ZoeDepth/releases/download/v1.0/ZoeD_M12_N.pt`

[4]`github.com/isl-org/ZoeDepth/releases/download/v1.0/ZoeD_M12_NK.pt`

[5]`node1.chrischoy.org/data/publications/fcgf/2019-08-19_06-17-41.pth`

[6]`node1.chrischoy.org/data/publications/fcgf/KITTI-v0.`
`3-ResUNetBN2C-conv1-5-nout32.pth`

exceeds $\tau_g$, we set its geometric feature to a zero vector. $\tau_g$ is set to 0.5m for indoor datasets and 5m for outdoor datasets.

### A.1.5 FEATURE FUSION

The RGB images and depth maps are resized to $512 \times 704$ for indoor datasets and $512 \times 1280$ for outdoor datasets. The output feature map of $\mathcal{U}_6$ is $16\times$ smaller than the original input. The Diffusion Feature maps for indoor and outdoor images are $32 \times 44$ and $32 \times 80$, respectively. We interpolate features on these feature maps to construct diffusion features.

### A.1.6 SOLVING RELATIVE TRANSFORMATION

Given two sets of fused features $F^I$ on RGB image $I$ and $F^D$ on depth map $D$, we conduct Nearest Neighbor (NN) feature matching with a mutual nearest check (Wang et al., 2022a) to find a set of correspondences between $I$ and $D$. Note that each pixel in $D$ corresponds to a 3D point in $P$. We thus obtain a set of pixel-to-point correspondences $\mathcal{C} = \{((p \in \mathbb{Z}^{+2}, q \in \mathbb{R}^3))\}$, where $p$ is a pixel coordinate in $I$ and $q$ is a point in point cloud $P$. Based on these correspondences, we employ the Kabsch algorithm to recover the I2P relative pose when we have Zoe-depth estimations. Otherwise, we use the Perspective-n-Point (PnP) method.

The Kabsch algorithm is formulated by

$$\{R, t\} = \underset{R \in SO(3), t \in \mathbb{R}^3}{\arg\min} \sum_{(p,q) \in \mathcal{C}} \|q - Rproj(p, d_p^Z, K^I) - t\|_2, \tag{3}$$

where $proj((u, v), d, K) = K^{-1}[u, v, 1]^T * d$ converts 2D pixel coordinates $(u, v) \in \mathbb{Z}^{+2}$ to a 3D point based on depth $d$ and intrinsic matrix $K$.

The PnP algorithm is formulated by

$$\{R, t\} = \underset{R \in SO(3), t \in \mathbb{R}^3}{\arg\min} \sum_{(p,q) \in \mathcal{C}} \|p - proj^{-1}(q, K^I, R, t)\|_2, \tag{4}$$

where $proj^{-1}(q, K^I, R, t)$ projects a 3D point $q$ to a 2D pixel coordinate according to the intrinsic matrix $K^I$ and I2P relative pose $(R, t)$ (Forsyth & Ponce, 2002).

To estimate the SE(3) relative transformation with PnP, we use PnP-RANSAC implemented in OpenCV (Bradski, 2000) with 50k iterations and the distance tolerance of 10.0. For Kabsch-based pose estimation, we adopt the Kabsch-RANSAC implemented in Open3D (Zhou et al., 2018) with 50k iterations and the distance tolerance of 0.2/4m for indoor/outdoor datasets.

## A.2 IMPLEMENTATION DETAILS OF BASELINE METHODS

**LCD** (Pham et al., 2020) is a cross-modality registration baseline published in AAAI 2020. We utilize the official codes and models. We extract LCD-2D/3D features on the same feature pixels/points as FreeReg for a fair comparison. LCD models are trained on the indoor 3DMatch (Zeng et al., 2017) dataset and almost fail on the outdoor Kitti-DC dataset.

**DeepI2P** (Li & Lee, 2021) is a cross-registration baseline published in CVPR 2021. We utilize the official codes and models pretrained on the Oxford RobotCar dataset (Maddern et al., 2017). However, it does not generalize well to both the indoor 3DMatch/ScanNet datasets and the outdoor Kitti-DC dataset.

**CorrI2P** (Ren et al., 2022) is a cross-modality baseline published in TCSVT 2022. The official codes are released but no pretrained model is available. We thus train it following the official guidance. The training process takes about 14 hours for 25 epoches. The trained model achieved an average rotation error of 7 degrees on their dataset. However, the model almost failed on our KITTI-DC test data because CorrI2P relies on the intensity information of the point cloud as input, which is not available in our test data and we set them to ones.

**SuperGlue** (Sarlin et al., 2020) (SG) is a widely adopted image-matching method. We employed the official implementation and indoor/outdoor models to estimate correspondences between an RGB

image and a depth image. A total of 1024 key points were detected on every image, and the correspondence confidence threshold was set at 0.01 to retain more correspondences to solve PnP.

**P2-Net** (Wang et al., 2021) is a I2P registration baseline published in ICCV 2021. However, the P2-Net fails to converge on the 3DMatch trainset as stated in 2D3D-Matr (Li et al., 2023).

**I2P-Matr** extracts features from RGB images and point clouds separately. Then, 2D3D-Matr subsequently utilizes the nearest neighborhood matcher with a mutual nearest check for feature matching. Same as (Li et al., 2023), we utilize ResNet (He et al., 2016) and FPN (Lin et al., 2017) to construct a siamese network as the feature extractor for both RGB images and depth maps projected from point clouds. We adopt CircleLoss (Sun et al., 2020; Li et al., 2023) to supervise the training, and the training strategy is consistent with that of 2D3D-Matr (Li et al., 2023). We trained I2P-Matr on the 3DMatch (Zeng et al., 2017) trainset for 20 epochs, utilizing a learning rate of 1e-3, and applying a decay factor of 0.98 after each epoch. For the test, we adopt the same data pre-processing strategies and evaluation settings as FreeReg.

**ControlNet**(Zhang & Agrawala, 2023)-**SuperGlue**(Sarlin et al., 2020) (CTL-SG) is a combined baseline. We adopt the same settings as Sec. A.1.2 for ControlNet.

## A.3 COMAPRISON WITH CONCURRENT WORK 2D3D-MATR

In this section, we provide a comparison with the concurrent work 2D3D-Matr (Li et al., 2023) on their evaluation dataset RGBD-Scene-v2. Since their codes have not been released yet, we report their results from their paper. **2D3D-Matr** (Li et al., 2023) is a concurrent work of FreeReg. 2D3D-Matr adopts a feature training methodology similar to I2P-Matr, and further learns to estimate correspondence construction based on a dense matcher similar to GeoTransformer (Qin et al., 2022), yielding better correspondences and transformation solutions.

**Experimental protocol.** Following (Li et al., 2023), we conducted evaluation on $scene\_11-14$ from the $RGBD-Scenes-v2$ dataset (Lai et al., 2014). The dataset preparation is identical to 2D3D-Matr. For evaluation, we adopt three metrics same as 2D3D-Matr: (1) *Inlier Ratio (IR)*, the ratio of pixel-to-point matches whose 3D distance is below 5cm over all constructed matches. (2) *Feature Matching Recall (FMR)*, the ratio of image-to-point-cloud pairs whose inlier ratio is above $10\%$. (3) *Registration Recall (RR)*, the ratio of image-point-cloud pairs whose RMSE between the point clouds transformed by the ground truth and the predicted transformation is below $\tau = 10cm$. For a fair comparison, we follow (Li et al., 2023) to adopt a traditional coarse-to-fine strategy similar to 2D3D-Matr to establish dense correspondences. Specifically, we initially utilized $k$-nearest neighbor mutual matcher on the diffusion features to construct coarse correspondences. Subsequently, we built dense correspondences on local patches of the established correspondences utilizing geometric features and nearest neighbor (NN) matcher. Note that FreeReg requires no training on the I2P task, whereas 2D3D-Matr was trained on scenes 1-8 of the RGBD-Scenes-v2 dataset.

**Results**. The results are shown in Table. 5. FreeReg achieves a $30.9\%$ IR close to 2D3D-Matr and a better registration quality than 2D3D-Matr with RR of $57.3\%$. We notice FreeReg with the Kabsch method for transformation estimation performs much worse results than PnP in this experimental setting because the estimated depth maps from Zoe-Depth have a large global scale difference from real depth values. More analysis about the Kabsch and PnP can be found in Sec. A.4.4.

## A.4 MORE ABLATIONS AND RESULTS

### A.4.1 USE OTHER LARGE VISION BACKBONES RATHER THAN DIFFUSION NETWORKS.

In FreeReg, we utilize pre-trained diffusion models, specifically Stable Diffusion and ControlNet, to establish cross-modality consistent features between images and point clouds. A possible alternative method could be employing some other pre-trained large-scale vision foundation backbones(VFB), such as Dino-v2 (Oquab et al., 2023) and CLIP (Radford et al., 2021), for cross-modality feature construction. However, these VFBs are designed and trained to process RGB images, while point clouds encode geometric information rather than RGB information, containing distinct modality differences from RGB images. Thus we neglect the modality differences between point clouds and images. Specifically, we first project the input point cloud into a depth map and expand it to three channels. Subsequently, we use Dino-v2 (Oquab et al., 2023), CLIP (Radford et al., 2021),

Table 5: Quantitative results on RGBD-Scene-v2

| Method | Scene-11 | Scene-12 | Scene-13 | Scene-14 | Mean |
|---|---|---|---|---|---|
| *Feature Matching Recall (%)* | | | | | |
| FCGF-2D3D | 11.1 | 30.4 | 51.5 | 15.5 | 27.1 |
| P2-Net | 48.6 | 65.7 | 82.5 | 41.6 | 59.6 |
| Predator-2D3D | 86.1 | 89.2 | 63.9 | 24.3 | 65.9 |
| 2D3D-Matr | **98.6** | **98.0** | 88.7 | **77.9** | **90.8** |
| FreeReg | 91.9 | 93.4 | **93.1** | 49.6 | 82.0 |
| *Inlier Ratio (%)* | | | | | |
| FCGF-2D3D | 6.8 | 8.5 | 11.8 | 5.4 | 8.1 |
| P2-Net | 9.7 | 12.8 | 17.0 | 9.3 | 12.2 |
| Predator-2D3D | 17.7 | 19.4 | 17.2 | 8.4 | 15.7 |
| 2D3D-Matr | 32.8 | 34.4 | **39.2** | **23.3** | **32.4** |
| FreeReg | **36.6** | **34.5** | 34.2 | 18.2 | 30.9 |
| *Registration Recall (%)* | | | | | |
| FCGF-2D3D | 26.4 | 41.2 | 37.1 | 16.8 | 30.4 |
| P2-Net | 40.3 | 40.2 | 41.2 | 31.9 | 38.4 |
| Predator-2D3D | 44.4 | 41.2 | 21.6 | 13.7 | 30.2 |
| 2D3D-Matr | 63.9 | 53.9 | **58.8** | 49.1 | 56.4 |
| FreeReg+Kabsch | 38.7 | 51.6 | 30.7 | 15.5 | 34.1 |
| FreeReg+PnP | **74.2** | **72.5** | 54.5 | 27.9 | **57.3** |

Table 6: Results on 3DMatch utilizing features from other VFBs

| Methods | FMR (%) | IR (%) | IN (#) | RR (%) |
|---|---|---|---|---|
| Dino-v2 (Oquab et al., 2023) | 64.3 | 13.0 | 8.8 | 9.8 |
| CLIP (Radford et al., 2021) | 21.9 | 3.3 | 2.4 | / |
| Dino-SLayer (Zhang et al., 2023) | 35.5 | 8.9 | 12.6 | 14.8 |
| ATTF (Zhang et al., 2023) | 41.7 | 10.9 | 15.5 | 17.4 |
| FreeReg (Ours) | **94.6** | **47.0** | **82.8** | **63.8** |

or ATTF (Zhang et al., 2023) to extract features from the depth map and input RGB for matching. In Table 6, we report the performance of features from other VFBs in I2P registration. The results indicate that existing visual models struggle to overcome significant modality differences between images and point clouds, achieving only less than $13\%$ IR and less than $20\%$ RR on 3DMatch.

### A.4.2 USING STABLE DIFFUSION TO EXTRACT DEPTH FEATURES

In FreeReg, we employ CN to extract diffusion features on depth maps. An alternative approach would be to directly feed the noised depth map to SD for diffusion feature extraction. In table 7, we observe that using CN boosts FreeReg by $2\times$ on IN and $14.0\%$ on RR than directly feeding it to SD.

Table 7: Extract depth diffusion features with ControlNet (CN) or not.

| CN | FMR(%) | IR(%) | IN(#) | RR(%) |
|---|---|---|---|---|
| | 83.2 | 28.4 | 30.3 | 19.2 |
| ✓ | 91.9 | 39.6 | 60.8 | 33.2 |

### A.4.3 RESULTS UTILIZING DIFFERENT DIFFUSION LAYER FEATURES

In table 8, we utilize features of different diffusion layers to construct cross-modality diffusion features. In combinations (a/b), it can be observed that employing the later layers indexed over 6 leads to a sharp performance drop. Combinations (c-j,u) combine different layers from the three groups mentioned in Sec.4.3.1 of the main paper, i.e., small group (with a size of $8 \times 11$, layers 0-2), medium group (with a size of $16 \times 22$, layers 3-5), and large group (with a size of $32 \times 44$, layers 6-8). These combinations yield similar registration performances. Specifically, the combination [2,4,6] achieves the optimal FMR, [0,3,6] achieves the best IR, and [0,5,6] achieves the highest RR albeit with a significantly lower IR and IN. Combinations (h-p) fuse features of different layers within the same group and combinations (q-t) utilize more layers. These combinations do not yield a performance improvement. We empirically select the layer combination [0,4,6] which exhibits ideal correspondence and registration performance. The layer selection could be improved by future works to automatically select good features.

Table 8: Results on bundlefusion-office0 utilizing different diffusion layer features.

| ID | Layer | Feature Dimension | FMR(%) | IR(%) | IN(#) | RR(%) |
|----|-------|-------------------|--------|-------|-------|-------|
| (a) | [0,4,7] | $384 \times 32 \times 44$ | 92.2 | 37.2 | 56.1 | 31.4 |
| (b) | [0,4,8] | $384 \times 32 \times 44$ | 92.2 | 36.6 | 51.8 | 32.7 |
| (c) | [1,4,6] | $384 \times 32 \times 44$ | 93.5 | 41.5 | 56.2 | 35.3 |
| (d) | [2,4,6] | $384 \times 32 \times 44$ | 95.4 | 41.7 | 56.8 | 34.6 |
| (e) | [0,3,6] | $384 \times 32 \times 44$ | 94.8 | 43.6 | 60.4 | 34.6 |
| (f) | [0,5,6] | $384 \times 32 \times 44$ | 93.5 | 40.7 | 53.6 | **36.6** |
| (g) | [1,3,6] | $384 \times 32 \times 44$ | 93.5 | 41.8 | 60.2 | 35.3 |
| (h) | [1,5,6] | $384 \times 32 \times 44$ | 92.2 | 40.9 | 53.6 | 32.0 |
| (i) | [2,3,6] | $384 \times 32 \times 44$ | 91.5 | 43.2 | 58.5 | 34.6 |
| (j) | [2,5,6] | $384 \times 32 \times 44$ | 94.8 | 41.0 | 54.0 | 34.0 |
| (k) | [0,1,6] | $384 \times 32 \times 44$ | 92.2 | 43.6 | 59.8 | 34.6 |
| (l) | [0,2,6] | $384 \times 32 \times 44$ | 94.8 | 44.2 | 61.0 | 33.3 |
| (m) | [1,2,6] | $384 \times 32 \times 44$ | 93.5 | 43.9 | 59.1 | 31.4 |
| (n) | [3,4,6] | $384 \times 32 \times 44$ | 93.5 | 40.5 | 57.9 | 35.3 |
| (o) | [3,5,6] | $384 \times 32 \times 44$ | 92.8 | 40.0 | 55.0 | 34.6 |
| (p) | [4,5,6] | $384 \times 32 \times 44$ | 94.8 | 37.2 | 46.4 | 34.0 |
| (q) | [0,1,2,6] | $512 \times 32 \times 44$ | 92.8 | 45.3 | 59.1 | 33.3 |
| (r) | [3,4,5,6] | $512 \times 32 \times 44$ | 92.8 | 40.4 | 53.0 | 34.0 |
| (s) | [0,1,4,6] | $512 \times 32 \times 44$ | 93.5 | 43.8 | 57.7 | 34.6 |
| (t) | [0,3,4,6] | $512 \times 32 \times 44$ | 92.2 | 43.3 | 59.6 | 35.9 |
| (u) | [0,4,6] | $384 \times 32 \times 44$ | 94.8 | 42.3 | 58.2 | 35.9 |

Table 9: Results of PnP and Kabsch algorithms.

| | | | | 3DMatch | | | | |
|---|---|---|---|---|---|---|---|---|
| | Zoe-ADE (m) | RE(°) | TE(m) | | | Registration Recall (%) | | |
| | | | | (5°,0.1m) | (10°,0.2m) | (15°,0.3m) | (20°,0.5m) | (25°,0.5m) |
| FreeReg + PnP | 0.30 | 78.4 | / | 9.4 | 22.3 | 31.4 | 40.0 | 40.2 |
| FreeReg + Kabsch | | 22.5 | 0.659 | 3.5 | 20.5 | 41.6 | 63.8 | 65.3 |
| | | | | ScanNet | | | | |
| | Zoe-ADE (m) | RE(°) | TE(m) | | | Registration Recall (%) | | |
| | | | | (5°,0.1m) | (10°,0.2m) | (15°,0.3m) | (20°,0.5m) | (25°,0.5m) |
| FreeReg + PnP | 0.23 | 51.9 | / | 11.7 | 32.9 | 46.2 | 57.6 | 57.9 |
| FreeReg + Kabsch | | 14.0 | 0.429 | 8.2 | 34.0 | 58.2 | 78.0 | 78.5 |
| | | | | Kitti-DC | | | | |
| | Zoe-ADE (m) | RE(°) | TE(m) | | | Registration Recall (%) | | |
| | | | | (3°,1m) | (5°,2m) | (7°,3m) | (10°,3m) | (10°,4m) |
| FreeReg + PnP | 3.40 | 22.3 | 3.150 | 39.5 | 57.6 | 67.0 | 70.5 | 75.1 |
| FreeReg + Kabsch | | 6.2 | 2.559 | 5.3 | 26.0 | 55.0 | 67.5 | 80.1 |

### A.4.4 COMPARISON BETWEEN PnP AND KABSCH

In this section, we focus on comparing PnP algorithm with the Kabsch algorithm. In Table 9, we report the performance of FreeReg at different RR thresholds using PnP and Kabsch algorithms. We also provide the average depth error (ADE) of Zoe-Depth estimation, the average rotation error (RE), and the translation error (TE) of the estimated I2P transformations on the four datasets.

It can be seen that under strict thresholds, PnP achieves a higher RR, whereas Kabsch experiences a sharp performance drop. Using the Kabsch method yields more correct transformations in larger RR thresholds than PnP, and has lower average transformation errors on all thresholds. The main reason is that the Zoe-Depth predictions are not absolutely accurate, especially in terms of scales. It has a 0.27m depth error in indoor datasets and a 3.4m depth error in outdoor scenes which leads to a bias in the Kabsch estimation.

For further analysis, we provide the recall under different rotation and translation errors on the three datasets in Fig. 7. It can be seen that on the ScanNet dataset with a relatively low estimate depth error for Zoe-Depth, the Kabsch algorithm estimates more accurate I2P registrations, which is similar to PnP even under strict thresholds. However, on 3Dmatch and KITTI-DC with larger depth errors for Zoe-Depth, the translation accuracy of Kabsch estimations is much lower. Nevertheless, Kabsch leverage Zoe-depth to constrain the transformation estimation thus successfully registering more I2P pairs under a loose RR threshold.

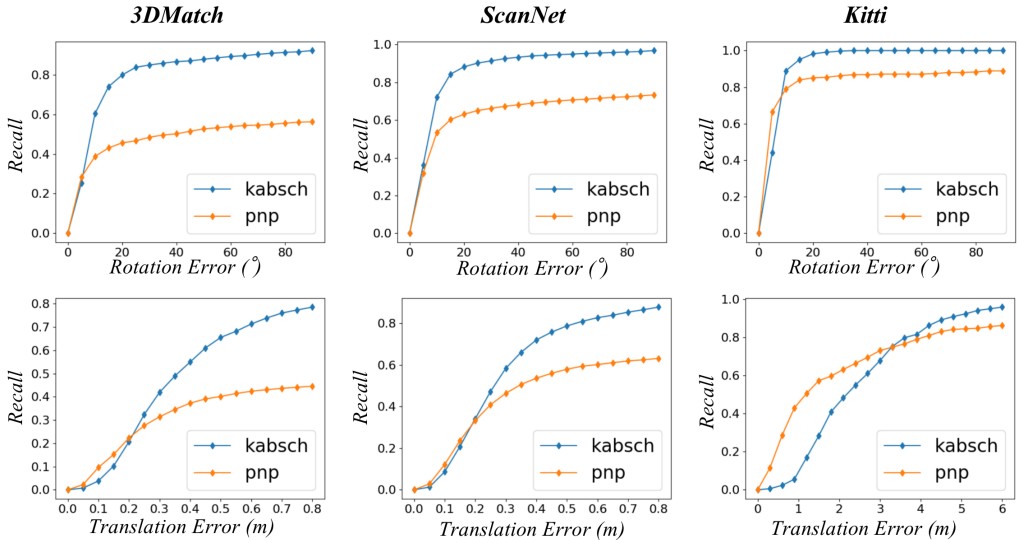

Figure 7: Rotation and translation errors on three benchmarks.

Table 10: *Mono-modality registration performance.* "RGB" means registering RGB images. "DPT" means 3D registration on point clouds recovered from depth maps.

| *Method* | *RGB* | *DPT* | **3DMatch** | | | | **ScanNet** | | | |
|---|---|---|---|---|---|---|---|---|---|---|
| | | | *FMR(%)* | *IR(%)* | *IN(#)* | *RR(%)* | *FMR(%)* | *IR(%)* | *IN(#)* | *RR(%)* |
| LCD | ✓ | | 94.6 | 48.9 | 101.0 | / | 99.7 | 53.0 | 140.0 | / |
| SG | ✓ | | **99.6** | **83.8** | 161.9 | / | **99.9** | **92.6** | 117.8 | / |
| I2P-Matr | ✓ | | 94.7 | 46.1 | 140.8 | / | 99.1 | 45.1 | 146.3 | / |
| FreeReg | ✓ | | 98.3 | 65.3 | **185.6** | 74.2 | 99.9 | 72.8 | 249.2 | 87.9 |
| LCD | | ✓ | 76.9 | 18.8 | 46.0 | 44.9 | 92.6 | 32.2 | 79.9 | 71.8 |
| FCGF | | ✓ | 95.4 | 54.5 | 121.3 | 84.7 | **99.8** | **75.5** | **230.8** | **95.3** |
| I2P-Matr | | ✓ | 96.1 | 43.9 | **140.4** | 80.9 | 98.7 | 55.3 | 174.3 | 91.6 |
| FreeReg | | ✓ | **96.4** | 55.7 | 131.8 | **87.0** | 99.5 | 73.8 | 202.5 | 94.7 |

### A.4.5 RESULTS ON MONO-MODALITY BENCHMARKS

FreeReg features are capable of performing mono-modality registration on RGB or 3D data. The qualitative results are given in Table 10. Given RGB image pairs, FreeReg employs Stable Diffusion (Rombach et al., 2022) to construct diffusion features in FreeReg-D and uses Zoe-Depth to estimate depth maps of query RGB images for further constructing geometric features for FreeReg. When given depth map pairs, FreeReg uses ControlNet (Zhang & Agrawala, 2023) to extract diffusion features for matching in FreeReg-D, and further combine them with Geometric Features extracted on depth maps in FreeReg.

**RGB Registration.** FreeReg (FreeReg) significantly outperforms fully-supervised cross-modality registration baseline LCD (Pham et al., 2020) and I2P-Matr by $18.1\%/23.5\%$ and $1.8 \times /1.5\times$ on IR and IN when registering RGB images. Compared with fully-supervised RGB registration baseline SuperGlue (Sarlin et al., 2020), features of FreeReg are distinctive enough to construct $1.6\times$ more inlier correspondences. However, FreeReg under-performs SuperGlue by $19.2\%$ on IR. This is mainly because SuperGlue applies a transformer for matching, whereas FreeReg constructs correspondences by a simple NN matcher. In terms of the final registration results, FreeReg is able to produce better registration results due to its utilization of Zoe-Depth.

**3D Registration.** For 3D registration on depth map pairs from 3DMatch and ScanNet, FreeReg significantly outperforms cross-modality baseline LCD (Pham et al., 2020) and I2P-Matr. In comparison with 3D-registration baseline FCGF (Choy et al., 2019), FreeReg has comparable results with better performance on the 3D Match dataset but worse results on the ScanNet dataset.

Table 11: FreeReg performances before or after fine-tuning (-Ft).

| | FMR (%) | IR (%) | IN (#) | RR (%) |
|---|---|---|---|---|
| | | 3DMatch | | |
| FreeReg | 94.6 | 47.0 | 82.8 | 63.8 |
| FreeReg-Ft | 95.8 (+1.2) | 53.7 (+6.7) | 95.4 (+12.6) | 70.1 (+6.3) |
| | | ScanNet | | |
| | FMR (%) | IR (%) | IN (#) | RR (%) |
| FreeReg | 98.5 | 56.8 | 114.4 | 78.0 |
| FreeReg-Ft | 98.9 (+0.4) | 62.4 (+5.6) | 126.1 (+11.7) | 81.4 (+3.4) |
| | | KITTI-DC | | |
| | FMR (%) | IR (%) | IN (#) | RR (%) |
| FreeReg | 99.7 | 58.3 | 132.9 | 70.5 |
| FreeReg-Ft | 100 (+0.3) | 62.8 (+4.5) | 156.1 (+23.2) | 81.9 (+11.4) |

Table 12: Comparison of 2D3D-Matr and FreeReg with fine-tuning.

| Method | Scene-11 | Scene-12 | Scene-13 | Scene-14 | Mean |
|---|---|---|---|---|---|
| | *Feature Matching Recall (%)* | | | | |
| 2D3D-Matr | **98.6** | **98.0** | 88.7 | **77.9** | **90.8** |
| FreeReg | 91.9 | 93.4 | 93.1 | 49.6 | 82.0 |
| FreeReg-Ft | 93.5 | 95.6 | **95.0** | 63.3 | 86.9 |
| | *Inlier Ratio (%)* | | | | |
| 2D3D-Matr | 32.8 | 34.4 | 39.2 | 23.3 | 32.4 |
| FreeReg | 36.6 | 34.5 | 34.2 | 18.2 | 30.9 |
| FreeReg-Ft | **40.3** | **40.9** | **46.2** | **25.9** | **38.3** |
| | *Registration Recall (%)* | | | | |
| 2D3D-Matr | 63.9 | 53.9 | 58.8 | **49.1** | 56.4 |
| FreeReg | 74.2 | 72.5 | 54.5 | 27.9 | 57.3 |
| FreeReg-Ft | **80.6** | **82.4** | **73.3** | 38.1 | **68.6** |

### A.4.6 MORE QUALITATIVE RESULTS.

We provide more qualitative results in Fig. 9. The estimated cross-modality correspondences can be utilized to warp local image patches onto the point cloud as illustrated in Fig. 10.

## A.5 MORE ANALYSIS ON FREEREG

### A.5.1 HOW TO FINE-TUNE FREEREG FOR IMPROVED PERFORMANCES?

We found that ControlNet's pre-training model is trained on depth maps from monocular depth estimation rather than depth data collected from sensors (Zhang & Agrawala, 2023). Then, we argue that if sensor-captured RGB-D data is available, ControlNet fine-tuning can significantly improve FreeReg's performance. We conducted depth-to-RGB ControlNet fine-tuning for $\sim 1$ **hour** using sensor-captured RGB-D data from the train-set of 3DMatch/ScanNet/KITTI-DC. Then, on 3DMatch/ScanNet/KITTT-DC test-set, FreeReg-Ft achieved IR improvements of $6.7\%/5.6\%/4.5\%$ and RR improvements of $6.3\%/3.4\%/11.4\%$ as shown in Table 11. Comparing with 2D3D-Matr in Table 12, FreeReg-Ft achieved a $5.9\%$ increase in IR and a $12.2\%$ increase in RR.

Table 13: FreeReg acceleration with fewer DDIM sampling iterations.

| DDIM-Iters (#) | FMR (%) | IR (%) | IN (#) | RR (%) | Time (s) |
|---|---|---|---|---|---|
| 5 | 93.6 | **47.8** | 72.1 | 62.4 | **4.7** |
| 10 | **95.0** | 46.6 | 83.0 | 63.5 | 6.4 |
| 15 | 94.5 | 47.1 | **82.4** | 63.0 | 8.1 |
| 20 (Ours) | 94.6 | 47.0 | 82.8 | **63.8** | 9.3 |

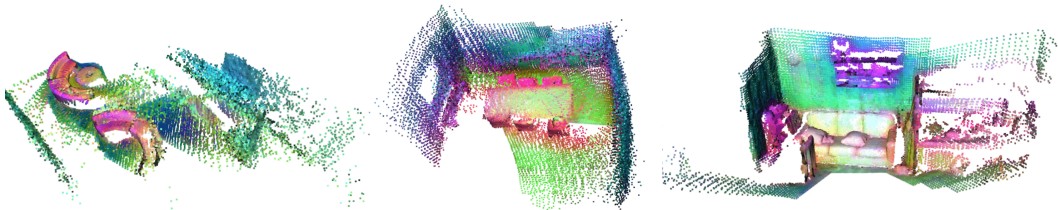

Figure 8: Semantic properties of FreeReg features.

### A.5.2 How to accelerate FreeReg?

Here we provide two insights for FreeReg speed-up.

**ControlNet acceleration.** The most time-consuming part of FreeReg registration is the sampling process of ControlNet. As described in Sec. A.1.2, we adopt DDIM Sampling strategy (Song et al., 2020a) with diffusion steps set to $range(begin = 1000, end = 0, step = -50)$ for ControlNet execution, i.e., 20 sampling iterations by defaults. We can enlarge the sampling step interval for fewer sampling iterations. As shown in table 13, when the number of the DDIM iterations is reduced to only 5 samples, FreeReg achieves a registration time reduction of $\sim 50\%$, with only a $1.4\%$ decrease in RR. This could be further optimized with the emergence of diffusion sampling speed-up strategies (Luo et al., 2023b).

**Knowledge distillation.** The features of FreeReg are directly derived from the pre-trained large model Stable Diffusion, providing powerful descriptive capabilities but is slow. We thus can employ FreeReg as a teacher model to guide the training of a lightweight but strong student image/point cloud feature extractor through knowledge distillation.

### A.5.3 Are FreeReg features suitable for other cross-modality tasks?

Besides cross-modality consistency, we argue that FreeReg features are semantic-aware. In Fig. 8, we present some visualizations of point cloud FreeReg features, revealing strong semantic relevance. This suggests the potential applicability of these features for downstream tasks requiring semantics.

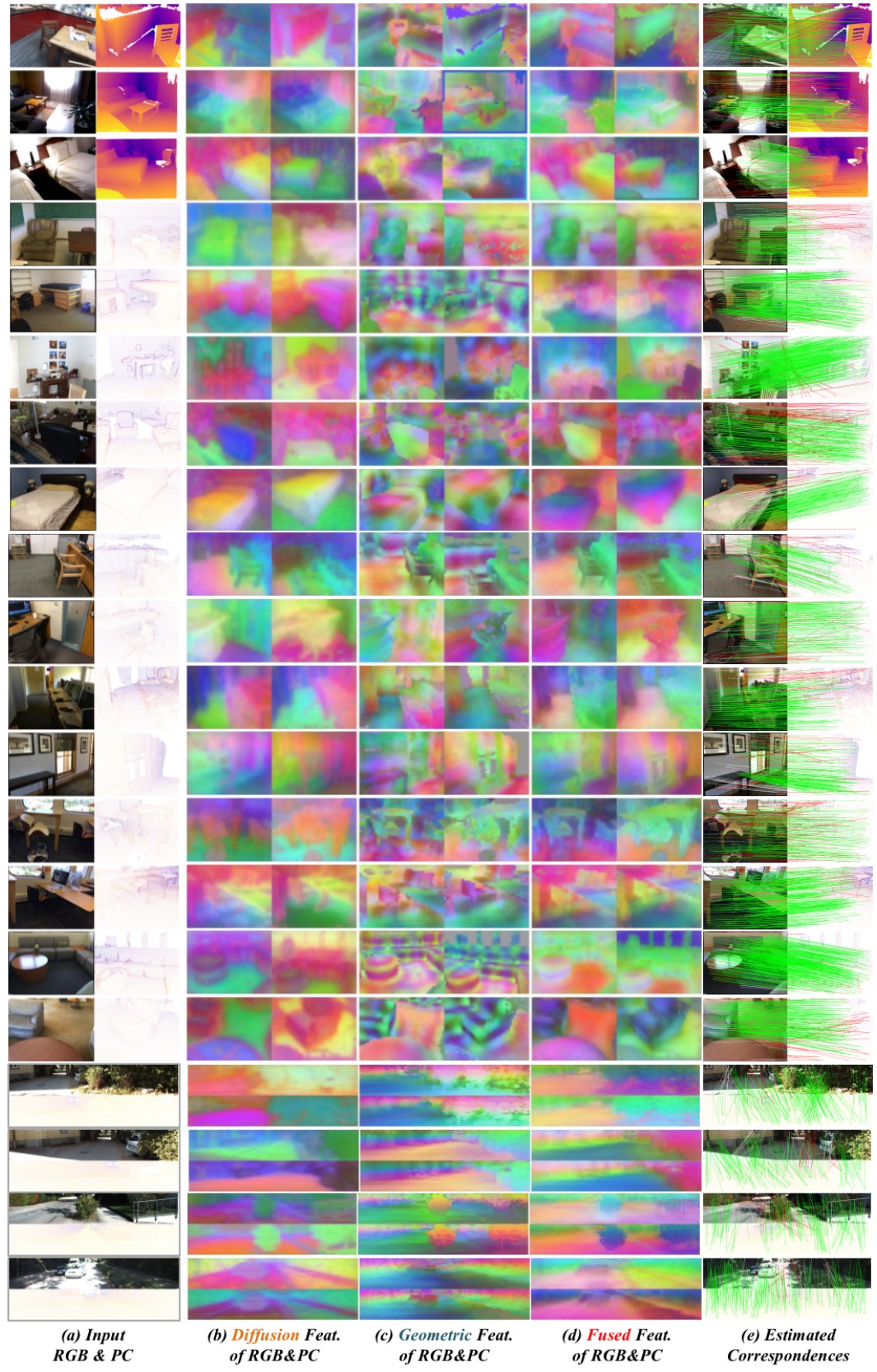

| (a) Input RGB & PC | (b) *Diffusion* Feat. of RGB&PC | (c) *Geometric* Feat. of RGB&PC | (d) *Fused* Feat. of RGB&PC | (e) Estimated Correspondences |
|---|---|---|---|---|

Figure 9: *Additional qualitative results.* (a) Input RGB images and depth maps for registration. (b/c/d) Diffusion / Geometric / Fused Feature maps of the input RGB images and depth maps. (e) Estimated correspondences.

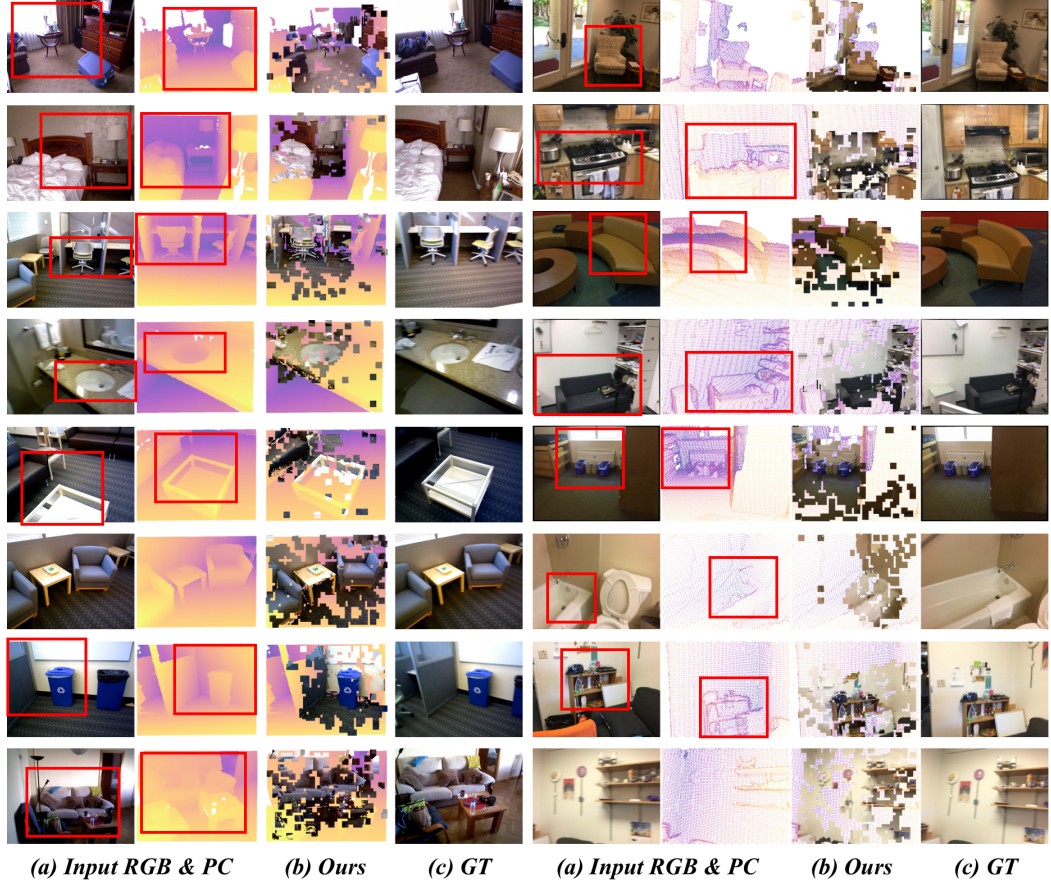

(a) Input RGB & PC    (b) Ours    (c) GT    (a) Input RGB & PC    (b) Ours    (c) GT

Figure 10: *Patch warping results.* Based on the estimated FreeReg correspondences, we warp the $32 \times 32$ local RGB patches to their estimated corresponding positions in the point cloud. (a)The input RGB and depth images. (b)The RGB patch warping results are based on the established correspondences of FreeReg. (c) The ground truth RGB image of the input depth map.

