# OpenReview forum: "FreeReg: Image-to-Point Cloud Registration Leveraging Pretrained Diffusion Models and Monocular Depth Estimators"
_ICLR.cc/2024/Conference — ICLR 2024 poster_

### Official Review · Reviewer_fhKS · 2023-10-30

**Soundness:** 2 fair
**Presentation:** 3 good
**Contribution:** 3 good
**Rating:** 6
**Confidence:** 3

**Summary:**

Overall, the core idea of the paper is interesting, which considers leveraging diffusion networks to achieve feature enhancing in the task of image-point-cloud registration.

**Strengths:**

Overall, the core idea of the paper is interesting, which considers leveraging diffusion networks to achieve feature enhancing in the task of image-point-cloud registration.

**Weaknesses:**

1. The idea of using diffusion networks is interesting, but the way to use diffusion networks is to some extent trivial. It is not well-motivated  why you choose to use diffusion networks instead of any other pretrained feature extractor, such as those vision foundation backbones (DINO, SAM, CLIP, ...)?

2. In Related Work (Image-to-point cloud registration), "In contrast, FreeReg does not require task-specific
training and finetuning and exhibits strong generalization ability to both indoor and outdoor scenes". It seems this description is not solid, as you feature extractors (diffusion networks and depth estimation networks) are already trained on some datasets.

3. The baseline comparison is a little bit weak. More recent and related works should be considered. And in Table S1, the proposed approach seems not solid outperform the counterpart 2D3D-Matr.
[1] Corri2p: Deep image-to-point cloud registration via dense correspondence. TCSVT 2022.
[2] EP2P-Loc: End-to-End 3D Point to 2D Pixel Localization for Large-Scale Visual Localization. ICCV 2023.
[3] CFI2P: Coarse-to-Fine Cross-Modal Correspondence Learning for Image-to-Point Cloud Registration. arXiv 2023.
[4] CoFiI2P: Image-to-Point Cloud Registration withCoarse-to-Fine Correspondences for Intelligent Driving. arXiv 2023.
[5] End-to-end 2D-3D Registration between Image and LiDAR Point Cloud for Vehicle Localization. arXiv 2023.


4. In Table 4, what is the performance under w=0 and w=1.0 ?

5. The run time speed/memory comparison with other models is missing.

**Questions:**

Please refer to Weaknesses.

---

> ### Author Response · Authors · 2023-11-19
> **Responses to Reviewer fhKS (part 1/2)**
>
> **We appreciate the reviewer's insightful comments and address all concerns below.**
>
> **Q1:** The motivation to use diffusion networks rather than other vision foundation backbones (VFB).
>
> **A1:** Existing VFBs are designed to process 2D RGB images and are not directly applicable to 3D point cloud data. Treating depth maps as RGB images and feeding them into these VFBs [1,2,3] for feature extraction and feature matching leads to poor performances as shown in Table 1 below. We have added this experiment in Sec.A.4.1 of the revised supplementary material.
>
> **Table 1: Results on 3DMatch utilizing features from other VFBs**
> | Method           | FMR (%) | IR (%) | IN (#) | RR (%) |
> |------------------|:-------:|:------:|:------:|:------:|
> | DINO-v2[1]       |  64.3   |  13.0  |  8.8   |  9.8   |
> | CLIP[2]          |  21.9   |   3.3  |   2.4  |    /   |
> | DINO-SLayer[3]   |  35.5   |  8.9   |  12.6  |  14.8  |
> | ATTF[3]          |  41.7   |  10.9  |  15.5  |  17.4  |
> | FreeReg          |  94.6   |  47.0  |  82.8  |  63.8  |
>
> [1] Oquab M, Darcet T, Moutakanni T, et al. Dinov2: Learning robust visual features without supervision[J]. arXiv preprint arXiv:2304.07193, 2023.
>
> [2] Radford A, Kim J W, Hallacy C, et al. Learning transferable visual models from natural language supervision[C]//International conference on machine learning. PMLR, 2021: 8748-8763.
>
> [3] Zhang J, Herrmann C, Hur J, et al. A Tale of Two Features: Stable Diffusion Complements DINO for Zero-Shot Semantic Correspondence[J]. arXiv preprint arXiv:2305.15347, 2023.
>
>
> **Q2:** ''FreeReg does not require task-specific training and finetuning and exhibits strong generalization ability to both indoor and outdoor scenes'' is not solid, as the feature extractors are already trained on some datasets.
>
> **A2:** We use "task-specific" to denote the training on the image-to-point cloud registration task. We agree that our pre-trained models are trained on other tasks. We make this clearer in the revised version by using "FreeReg does not need any training or fine-tuning on I2P registration task."
>
> **Q3.1:** FreeReg seems not solid outperform the counterpart 2D3D-Matr.
>
> **A3.1:** FreeReg already achieves 0.9% higher RR than 2D3D-Matr using only pre-trained models.
> Note 2D3D-Matr is fully supervised on the image-to-point cloud registration task and is a concurrent work to our submission.
> In Sec.A.5.1 of the revised supplementary material, we show that we fine-tune ControlNet on the image generation task (rather than the I2P registration task) for 1 hour and the resulted model outperforms 2D3D-Matr more significantly as shown in Table 2 below.
>
> **Table 2: Comparison of 2D3D-Matr and FreeReg(-Ft).**
> | Method     | FMR (%) | IR (%) | RR (%) |
> |------------|---------|--------|--------|
> | 2D3D-Matr  |   90.8  |  32.4  |  56.4  |
> | FreeReg    |   82.0  |  30.9  |  57.3  |
> | FreeReg-Ft |   86.9  |  38.3  |  68.6  |
>
>
> **Q3.2:** Comparision with additional baselines including CorrI2P, EP2P-Loc, C2FI2P,  E2E-I2P, and CoFiI2P.
>
> **A3.2:** Many thanks for your recommendation. EP2P-Loc (ICCV 2023), C2FI2P (arXiv), E2E-I2P (arXiv), and CoFiI2P (arXiv) are concurrent works to FreeReg and have not been open-sourced. We have added discussions of these concurrent works in Sec.2 of the revised main paper.
>
> CorrI2P [4] is the only open-sourced work and we tried to compare FreeReg with it. The official code of CorrI2P is released but no pretrained model is available. We thus train it on the provided training set. The trained model achieved an average rotation error of 7 degrees on their provided test dataset. However, the model almost fails on our KITTI-DC test data because CorrI2P relies on the intensity information of the point cloud as input, which is not available in our test data. We add such a discussion in Sec.A.2 of the revised supplementary material.
>
> [4] Ren S, Zeng Y, Hou J, et al. Corri2p: Deep image-to-point cloud registration via dense correspondence[J]. IEEE Transactions on Circuits and Systems for Video Technology, 2022, 33(3): 1198-1208.
>
>
> **Q4:** FreeReg performances under $w$=0 and $w$=1 in Table 4.
>
> **A4:** In our submission, FreeReg-D or FreeReg-G means $w=1$ or $w=0$ for feature matching respectively, for which we have reported their performances using PnP or Kabsch in Table 1.
> The performances with $w=1$ and $w=0$ using Kabsch are reported in Table 3 below and have been added to Table 4 of the revised main paper.
>
> **Table 3: FreeReg performances with w = 0 or 1.**
>
> |        $w$           |       FMR (%)       | IR (%) | IN (#) | RR (%) |
> |:-----------------:|:-------------------:|:------:|:------:|:------:|
> | 1 |        91.9         |  39.6  |  60.8  |  52.6  |
> | 0 |        90.7         |  31.4  |  49.4  |  50.4  |
> | 0.5 (Ours) |        94.6         |  47.0  |  82.8  |  63.8  |

---

> > ### Author Response · Authors · 2023-11-19
> > **Responses to Reviewer fhKS (part 2/2)**
> >
> > **Q5:** The run time speed/memory comparison with other models is missing.
> >
> > **A5:** Many thanks for your comment.
> >
> > 1. **Runtime and memory analysis.** In Table 4 below, we report the runtime and memory usage of the baselines and FreeReg for a single I2P registration on the 3DMatch dataset. LCD and SuperGlue achieved the shortest runtime but almost fail. These analyses have been added in Sec.4.4 of the revision.
> >
> >     **Table 4: Time and memory consumption of baselines for a single I2P registration.**
> >     | Methods   | FMR (%) | IR (%) | IN (#) | RR (%) | Time (s) | Memory (G) |
> >     |-----------|:-------:|:------:|:------:|:------:|:--------:|:----------:|
> >     | LCD       |  40.1   |  35.1  |  4.3   |    /   |   0.6    |    3.5     |
> >     | SG        |  50.3   |  11.1  |  3.1   |  1.8   |   0.6    |    2.3     |
> >     | CN+SG     |  64.7   |  18.4  |  10.9  |  6.5   |   6.4    |    11.6    |
> >     | I2D-Matr  |  90.6   |  24.9  |  49.0  |  28.2  |   1.7    |    2.7     |
> >     | FreeReg   |  94.6   |  47.0  |  82.8  |  63.8  |   9.3    |    12.7    |
> >
> > 2. **FreeReg acceleration.** We also provided possible direction to speed up FreeReg in Sec.A.5.2 of the revised supplementary material. The most time-consuming part of FreeReg registration is the forward sampling process of ControlNet so we can enlarge the sampling step interval for fewer sampling iterations for acceleration. As shown in Table 5 below, with only 5 sampling iterations, FreeReg achieves a registration time reduction of $\sim 50\%$, with only a $1.4\%$ decrease in RR.
> > This might be further optimized with the emergence of diffusion sampling speed-up strategies [5].
> >
> >     **Table 5: FreeReg acceleration with fewer DDIM sampling iterations.**
> >     | DDIM-Iters (#) | FMR (%) | IR (%) | IN (#) | RR (%) | Time (s) |
> >     |----------------|:-------:|:------:|:------:|:------:|:--------:|
> >     | 5              |  93.6   |  47.8  |  72.1  |  62.4  |   4.7    |
> >     | 10             |  95.0   |  46.6  |  83.0  |  63.5  |   6.4    |
> >     | 15             |  94.5   |  47.1  |  82.4  |  63.0  |   8.1    |
> >     | 20 (FreeReg)   |  94.6   |  47.0  |  82.8  |  63.8  |   9.3    |
> >
> >
> > [5] Luo S, Tan Y, Huang L, et al. Latent Consistency Models: Synthesizing High-Resolution Images with Few-Step Inference[J]. arXiv preprint arXiv:2310.04378, 2023.

---

> > ### Comment · Reviewer_fhKS · 2023-11-19
> > **Response to authors**
> >
> > Many thanks for the authors' responses ($\bf{A2}$ - $\bf{A5}$), which have answered my questions.
> >
> > For $\bf{A1}$, I am still conerned about the motivation. A1 is just an illustration of the results, instead of a motivation. What we readers really want is the deeper root reason why the diffusion net is working better than others. For example, is it because different VFBs use different pre-training proxy task and a suitable proxy task is more benificial to the down-stream registration task?
> >
> > Note that this kind of motivation is exactly the core of this whole paper, which determines whether this paper is a $\bf{great}$ "motivating and insightful" paper or just a $\bf{normal}$ "trick and try" paper. I hope authors could bring more thinkings.
> >
> > Also, for "Existing VFBs are designed to process 2D RGB images and are not directly applicable to 3D point cloud data", this is not a solid answer, as I can \
> > (1) use powerful 3D backbones, or \
> > (2) project 3D to 2D view to use 2D backbones (the same as this work).

---

> > > ### Author Response · Authors · 2023-11-19
> > > **Response to Reviewer fhKS**
> > >
> > > Many thanks for your further comments. Our motivations of using diffusion models rather than other vision foundation backbones are stated as follows
> > >
> > > 1. It is difficult to train a powerful 3D foundation model due to the scarcity of 3D data. A 3D dataset usually only has millions or thousands 3D data and most of them are object-level, which are much smaller than 2D image datasets of billion-level size. Thus, there are currently no 3D large models as we know can be effectively used in our scene-level image-to-point cloud registration.
> > >
> > > 2. Existing 2D VFBs are trained to process RGB images. Directly treating the depth map as an RGB image to utilize 2D VFB for feature extraction and matching does not work as shown by our previous replies.
> > >
> > > 3. We observed that ControlNet has the ability to generate 2D RGB images from depth maps, which is the only VFB that connects 2D images with 3D data to the best of our knowledge. Thus, we choose the diffusion model to align the features of images with the features of point clouds for cross-modality feature matching. The ControlNet is trained to generate images from depth maps, which forces them to learn similarities and differences between two modalities and thus possesses a strong ability for cross-modality feature matching.

---

> > > > ### Comment · Reviewer_fhKS · 2023-11-19
> > > > **Response to authors**
> > > >
> > > > Thanks for the timely replies by the authors.
> > > >
> > > > Even though the answer is still a shallow reason, this work overall is still an interesting try and has its value in cross-modal related-tasks.
> > > >
> > > > Therefore, I would like to raise my Rating.

---

> > > > > ### Author Response · Authors · 2023-11-19
> > > > > **Response to Reviewer fhKS**
> > > > >
> > > > > Thank you for the discussion! We really appreciate your efforts in reading our paper and helping us improve the paper!

---

### Official Review · Reviewer_zNJ2 · 2023-10-31

**Soundness:** 3 good
**Presentation:** 3 good
**Contribution:** 3 good
**Rating:** 8
**Confidence:** 2

**Summary:**

This paper introduces a image-to-point cloud registration framework. The key idea is to generate RGB image from point cloud and reconstruct depth image from RGB so that correspondences can be established between images of the same modality. Though the image generation of both directions are well studied, a naive implementation does not work well. For this reason, the authors first generate depth image from point cloud and then use intermediate feature maps in the depth-to-image ControlNet to establish semantic correspondence with the original image. At the same time, a depth map is generated from the original image and local geometric features extracted from the depth map are combined with the semantic features for better correspondences. Experiments are conducted on three datasets, including both indoor and outdoor scenes.

**Strengths:**

1.The idea of first generating images and point clouds from the other modality and then find correspondence in the same modality is interesting and the authors find practical ways to implement this idea.

2.The performance is promising even without training on the target task with ground-truth correspondence.

3. The paper is well written and the adequate ablation studies are conducted.

**Weaknesses:**

1. As mentioned by the author, inference speed is a limit of the proposed method and 11s per image is quite slow.  I hope that the author can provide their thoughts for further improvement of speed.

2. Another limitation is that the performance is only comparable with the concurrent work 2D3D-MATR on the dataset RGBD-Scene-v2, while  I think that this is not a big problem and the proposed method has its own value.

3. The residual rotation error seems quite large. I'd like to know what's the initial rotation error before registration?

**Questions:**

1. Is there any way that can are readily to be tried for speed improvement?

2. The residual rotation error is quite high. From a practical point of view, is the rotation error of 10 or 20 degrees a good threshold for recall?  What's the requirements of rotation accuracy in different application areas?

3. The authors "analyze" the limitation of straightforward/direct implementations in the 4th and 5th paragraph of Introduction, but did not provide any experimental results to support supporting the conclusion.  For the task of this paper and the fusion of two kinds of features, these straightforward may also work well.

---

> ### Author Response · Authors · 2023-11-19
> **Responses to Reviewer zNJ2**
>
> **We appreciate the reviewer's insightful comments and our responses are listed below.**
>
> **Q1:** Only comparable performances with the concurrent work 2D3D-Matr.
>
> **A1:** In Sec. A.5.1 of the revised supplementary material, we fine-tune ControlNet on the image generation task (rather than the difficult I2P registration task) for only 1 hour and the resulted model outperforms 2D3D-Matr more significantly as shown in Table 1 below.
>
> **Table 1: Comparison of 2D3D-Matr and FreeReg(-Ft).**
> | Method     | FMR (%) | IR (%) | RR (%) |
> |------------|---------|--------|--------|
> | 2D3D-Matr  |   90.8  |  32.4  |  56.4  |
> | FreeReg    |   82.0  |  30.9  |  57.3  |
> | FreeReg-Ft |   86.9  |  38.3  |  68.6  |
>
>
> **Q2:** How to accelerate FreeReg?
>
> **A2:** Many thanks for your inspiring comments. There are two possible ways to accelerate FreeReg.
>
> 1. **DDIM sampling speed-up.** The most time-consuming part of FreeReg registration is the forward sampling process of ControlNet and we can enlarge the sampling step interval for fewer sampling iterations for acceleration. As shown in Table 2 below, with only 5 sampling iterations, FreeReg achieves a registration time reduction of $\sim 50\%$, with only a $1.4\%$ decrease in RR. This might be further optimized with the emergence of diffusion sampling speed-up strategies[2].
>
>     **Table 2: FreeReg acceleration with fewer DDIM [1] sampling iterations.**
>     | DDIM-Iters (#) | FMR (%) | IR (%) | IN (#) | RR (%) | Time (s) |
>     |----------------|:-------:|:------:|:------:|:------:|:--------:|
>     | 5              |  93.6   |  47.8  |  72.1  |  62.4  |   4.7    |
>     | 10             |  95.0   |  46.6  |  83.0  |  63.5  |   6.4    |
>     | 15             |  94.5   |  47.1  |  82.4  |  63.0  |   8.1    |
>     | 20 (FreeReg)   |  94.6   |  47.0  |  82.8  |  63.8  |   9.3    |
>
> 2. **Knowledge distillation.** We can use FreeReg as a teacher network to guide the feature estimation of a lightweight student image/point cloud feature extractor. An analysis has been added in Sec.A.5.2 of the revised supplementary material.
>
> [1] Song J, Meng C, Ermon S. Denoising diffusion implicit models[J]. arXiv preprint arXiv:2010.02502, 2020.
>
> [2] Luo S, Tan Y, Huang L, et al. Latent Consistency Models: Synthesizing High-Resolution Images with Few-Step Inference[J]. arXiv preprint arXiv:2310.04378, 2023.
>
> **Q3.1:** What's the initial rotation error before registration?
>
> **A3.1:** The average initial rotation error between unposed images and posed point clouds is 115.4 degrees and 68.3 degrees on 3Dmatch and KITTI-DC, respectively.
>
> **Q3.2:** Is the rotation error of 10 or 20 degrees a good threshold for recall?
>
> **A3.2:** We agree that 10/20 degrees is a large rotation threshold and we select it based on the following considerations:
> 1. **Why 10/20 degrees:** The rotation threshold is typically set to 15/5 degree for the indoor/outdoor point cloud registration task[3]. As a cross-modality task,  Image-to-point cloud registration is more difficult. Thus, we use a looser threshold at 20 degrees for indoor I2P registration and a 10-degree threshold for outdoor registration, which is the same as DeepI2P.
> 2. **Performance under different thresholds:** We also report the performance of FreeReg under different thresholds in Fig. 1 and Table 5 in the revised supplementary material. On more than 50% data, our method achieves a rotation error less than 10/5 degrees for indoor/outdoor data.
>
> [3] Choy C, Park J, Koltun V. Fully convolutional geometric features[C]//Proceedings of the IEEE/CVF international conference on computer vision. 2019: 8958-8966.
>
> **Q3.3:** Is a rotation error of 10 or 20 degrees useful in practice?
>
> **A3.3:** Yes. 10/20 degree rotation accuracy can support applications like positioning and navigation in a mall, which do not require very high accuracy. We believe improving FreeReg's accuracy will enlarge the application scope of I2P registration.
>
> **Q4:** Experimental results of straightforward/direct implementations in the 4th and 5th paragraph of Introduction.
>
> **A4:** In the fourth paragraph, we analyzed that naively using ControlNet to directly transform input point clouds into the image modality failed to reliably match with input images. This corresponds to ControlNet+SuperGlue(CN+SG) in Sec.4. In the fifth paragraph, we discussed using Zoe-Depth to transform input images into the point cloud and matching between the point clouds. This corresponds to Zoe-Depth+FCGF, i.e., FreeReg-G in Sec.4. The experimental results indicate that both baselines perform worse than FreeReg. We have improved the introduction about these two baselines and recall the above discussions in Section 4.1 to make it clearer.

---

> > ### Comment · Reviewer_zNJ2 · 2023-11-20
> >
> > Thanks for the author's response and new results. My questions have been addressed and I keep my rating.

---

> > > ### Author Response · Authors · 2023-11-20
> > > **Responses to Reviewer zNJ2**
> > >
> > > We really appreciate your efforts in reviewing our paper and the insightful comments that help us improve the paper!

---

### Official Review · Reviewer_BaLi · 2023-11-01

**Soundness:** 2 fair
**Presentation:** 3 good
**Contribution:** 2 fair
**Rating:** 6
**Confidence:** 4

**Summary:**

The paper studies the image-to-point-cloud registration problem. The idea is to utilize a diffusion model and ControlNet to generate diffusion features from input point cloud. For images and point clouds, the final features used for matching are composed of both a diffusion part and a geometric part in a weighted average fashion. The latter is extracted from FCGF to serve as geometric features. The pixel-to-point correspondences are then obtained by a NN matching with mutual check. Experiments show the empirical improvements in I2P matching in several benchmark datasets.

**Strengths:**

- High-quality writing: The content is well-written, with a focus on clarity, coherence, and precision.

- Improved results over baselines: The results outperform standard models, demonstrating significant enhancements in performance and accuracy.

- Efficient feature distillation and cross-modality matching: Large model features are distilled effectively, facilitating feature matching across different modes for improved system performance.

**Weaknesses:**

- The paper primarily focuses on the design of an improved feature that serves as a unifying element for both the image and depth map domains. While this feature has shown remarkable efficacy within this specific context, it may not be directly transferable to other cross-modality problems. Adapting it to different cross-modality tasks would necessitate careful and tailored design to ensure its successful application.

- Efficiency is indeed a concern as pointed out in the limitation section, as feature extraction via stable diffusion and ControlNet is a costly computation.

**Questions:**

1. Can the method work with only diffusion features from RGB and point clouds? e.g. Can the weighting between F_d and F_g be either 1 or 0?

---

> ### Author Response · Authors · 2023-11-19
> **Responses to Reviewer BaLi**
>
> **We appreciate the reviewer's encouraging comments and make responses below.**
>
> **Q1:** Use FreeReg features in other cross-modality problems.
>
> **A1:** Many thanks for your insightful comments. In Sec.A.5.3 of the revised supplementary material, we show that the FreeReg features have strong semantic consistency, which suggests the potential ability of these features for other semantic tasks.
>
> **Q2:** How to accelerate FreeReg?
>
> **A2:** There are two possible ways to accelerate FreeReg.
>
> 1. **DDIM sampling speed-up.** The most time-consuming part of FreeReg registration is the forward sampling process of ControlNet and we can enlarge the sampling step interval for fewer sampling iterations for acceleration. As shown in Table 1 below, with only 5 sampling iterations, FreeReg achieves a registration time reduction of $\sim 50\%$, with only a $1.4\%$ decrease in RR. This might be further optimized with the emergence of diffusion sampling speed-up strategies[2].
>
>     **Table 1: FreeReg acceleration with fewer DDIM [1] sampling iterations.**
>     | DDIM-Iters (#) | FMR (%) | IR (%) | IN (#) | RR (%) | Time (s) |
>     |----------------|:-------:|:------:|:------:|:------:|:--------:|
>     | 5              |  93.6   |  47.8  |  72.1  |  62.4  |   4.7    |
>     | 10             |  95.0   |  46.6  |  83.0  |  63.5  |   6.4    |
>     | 15             |  94.5   |  47.1  |  82.4  |  63.0  |   8.1    |
>     | 20 (FreeReg)   |  94.6   |  47.0  |  82.8  |  63.8  |   9.3    |
>
> 2. **Knowledge distillation.** We can use FreeReg as a teacher network to guide the feature estimation of a lightweight student image/point cloud feature extractor. An analysis have been added in Sec.A.5.2 of the revised supplementary material.
>
> [1] Song J, Meng C, Ermon S. Denoising diffusion implicit models[J]. arXiv preprint arXiv:2010.02502, 2020.
>
> [2] Luo S, Tan Y, Huang L, et al. Latent Consistency Models: Synthesizing High-Resolution Images with Few-Step Inference[J]. arXiv preprint arXiv:2310.04378, 2023.
>
> **Q3:** Can FreeReg work with only diffusion or geometric features by setting $w$ to 1 or 0.
>
> **A3:**
> In our submission, FreeReg-D or FreeReg-G means $w=1$ or $w=0$ for feature matching respectively, for which we have reported their performances using PnP or Kabsch in Table 1.
> The performances with $w=1$ and $w=0$ using Kabsch are reported in Table 2 below and have been added to Table 4 of the revised main paper.
>
> **Table 2: FreeReg performances with w = 0 or 1.**
>
> |        $w$           |       FMR (%)       | IR (%) | IN (#) | RR (%) |
> |:-----------------:|:-------------------:|:------:|:------:|:------:|
> | 1 |        91.9         |  39.6  |  60.8  |  52.6  |
> | 0 |        90.7         |  31.4  |  49.4  |  50.4  |
> | 0.5 (Ours) |        94.6         |  47.0  |  82.8  |  63.8  |

---

### Author Response · Authors · 2023-11-19
**Overall Response from Authors**

We sincerely thank the reviewers for their thoughtful feedback. We are quite encouraged that they found our work to be interesting (fhKS,zNJ2), promising (zNJ2), effective (BaLi), practical (zNJ2), having its own value (zNJ2), and achieving SoTA performances (BaLi). After carefully improving the quality of our submission, we present here a revised main paper and supplementary materials, and modifications have been highlighted in brown.

Changes in the main paper include:
- Qualitative comparison with additional baselines in Sec.2.
- FreeReg performances with $w$ set to 0 and 1 in Table 4.
- Time and memory discussion in Sec.4.4.
- Improved writing:
  - "without any task-specific training" $\to$ "without any training on the I2P registration task"
  - "FreeReg-D" $\to$ "FreeReg-D (i.e. $w$ = 1)", "FreeReg-G" $\to$ "FreeReg-G (i.e. $w$ = 0)" in Sec.4.1

More changes in the revised supplementary material include:
- A performance discussion of the baseline CorrI2P in Sec.A.2.
- Results on using other pre-trained visual backbones rather than diffusion networks in Sec.A.4.1.
- Results of fine-tuning FreeReg for improved performances in Sec.A.5.1.
- Results and discussion of how to accelerate FreeReg in Sec.A.5.2.
- Discussion of using FreeReg features in other cross-modality tasks in Sec.A.5.3.

---

### Meta-Review · Area_Chair_TsM1 · 2023-12-05

**Metareview:**

This paper presents a novel image-to-point cloud registration method. Firstly, the authors unified the modality between images and point clouds by pretrained large-scale models. Secondly, the authors established robust correspondence within the same modality. Then they extracted geometric features on depth maps produced by the monocular depth estimator and matched these geometric features. Experimental results confirm that the proposed method outperforms existing state-of-the-arts.

The paper receives two “marginally above the acceptance threshold” ratings and one “accept, good paper” rating.

Reviewer BaLi gives “marginally above the acceptance threshold”. Reviewer BaLi thinks this paper is well-written.  Reviewer BaLi is satisfied with the experimental results. But also, Reviewer BaLi suggests adapting the proposed method to other cross-modality tasks.  Reviewer BaLi also raises concerns about the efficiency of the proposed method. The authors gave their rebuttals.  Reviewer BaLi keeps the initial rating.

Reviewer zNJ2 gives “accept, good paper”.  Reviewer zNJ2 thinks the idea is interesting.  Reviewer zNJ2 is also  satisfied with the performance of the proposed method. Reviewer zNJ2 thinks this paper is well-written. But also,  Reviewer zNJ2 raises concerns about the efficiency of the proposed method. Reviewer zNJ2 thinks the residual rotation error is large. The authors gave their rebuttals. After reading them, Reviewer zNJ2  has no further questions.

Reviewer fhKS gives “marginally above the acceptance threshold”. Reviewer fhKS thinks the core idea is interesting. But also, Reviewer fhKS thinks the way to use diffusion networks is a bit trivial. Reviewer fhKS also thinks the baseline method is a little bit weak.  Reviewer fhKS wants to see the run time speed/memory comparison with other models. The authors gave their rebuttals. After reading them, Reviewer fhKS raises the rating.

Therefore, based on the reviewers’ comments, the paper can be accepted by ICLR.

**Justification For Why Not Higher Score:**

All reviewers raise concerns about the efficiency of the proposed method. And the residual rotation error is still large. The motivation of this paper is not very clear.

**Justification For Why Not Lower Score:**

The core idea of this paper is novel and interesting. The proposed method outperforms existing state-of-the-arts. This paper is well-written.

---

### Decision · Program_Chairs · 2024-01-16

Accept (poster)